# Disjoint Generation of Synthetic Data

**Anton Danholt Lautrup**                                          *lautrup@imada.sdu.dk*
*Department of Mathematics and Computer Science*
*University of Southern Denmark*

**Muhammad Rajabinasab**                                    *rajabinasab@imada.sdu.dk*
*Department of Mathematics and Computer Science*
*University of Southern Denmark*

**Tobias Hyrup**                                                    *hyrup@imada.sdu.dk*
*Department of Mathematics and Computer Science*
*University of Southern Denmark*

**Arthur Zimek**                                                    *zimek@imada.sdu.dk*
*Department of Mathematics and Computer Science*
*University of Southern Denmark*

**Peter Schneider-Kamp**                                        *petersk@imada.sdu.dk*
*Department of Mathematics and Computer Science*
*University of Southern Denmark*

**Reviewed on OpenReview:** *https://openreview.net/forum?id=LSzXkAWBKI*

## Abstract

We propose a new framework for generating tabular synthetic datasets via disjoint generative models. In this paradigm, a dataset is partitioned into disjoint subsets that are supplied to separate instances of generative models. The results are then combined post hoc by a joining operation that works in the absence of common variables/identifiers. The success of the framework is demonstrated through several case studies and examples on tabular data that help illuminate some of the design choices that one may make. The advantages achieved by the disjoint generation include: i) An observed increase in the empirical measurement of privacy. ii) Increased computational feasibility of certain model types. iii) Ability to generate synthetic data using a mixture of different generative models. Specifically, mixed-model synthesis bridges the gap between privacy and utility performance, providing highly competitive performance on Accuracy and Area Under the Curve for downstream tasks while significantly lowering the empirical re-identification risk.

## 1 Introduction

A common strategy for solving difficult tasks is to "divide and conquer", i.e., to break the task into smaller, manageable sub-problems which are solved individually and then combined post hoc into a solution for the original problem. This paradigm can be applied effectively (and often recursively) to many branches of computer science (e.g., Hoare (1961); Karatsuba & Ofman (1962); Cooley & Tukey (1965)), but remains to be explored for generative modelling. Motivated by this gap, we propose a new generative model procedure for synthetic data generation in the tabular regime using partitioning.

In the proposed *Disjoint Generative Models (DGMs)* framework, training data are partitioned column-wise into disjoint subsets of variables, these are then used to train separate generative models before a joining operation combines the independent outputs. Disjoint generation challenges the notion that using all available data together leads to a superior outcome. While learning the overall distribution is certainly

useful for the utility of synthetic data, some models struggle to reliably capture high-dimensional structures or overfit to the detriment of privacy. DGMs enable specifying partitions based on the strengths and weaknesses of the included models, mixed model generation, multi-modal data, and using the joining operation and its control parameters to balance utility and empirical privacy effectively.

This framework can be used with many types of models, datasets, and joining algorithms. In this work, we focus primarily on the special case where the joining operation is done by repeatedly querying a validation model with candidate joins. Additionally, we restrict the experimental treatment to tabular data to enable clearer evaluation. We establish the framework through various case studies and results, and propose future directions to be explored. We make an easy-to-use and extendable implementation of disjoint synthetic data generation for the community to use and iterate upon.[1]

## 2 Background

Realistic synthetic data is artificially generated proxies for actual data (e.g., patient records, consumer profiles), following the same statistical distributions and multivariate relationships without copying individual records (Rankin et al., 2020). This makes synthetic data attractive for privacy protection, data amplification, and fairness-oriented augmentation (Hernandez et al., 2022; Hyrup et al., 2025; van Breugel et al., 2021; Fonseca & Bacao, 2023; Lautrup et al., 2024a). In particular, the generation of tabular synthetic data has emerged as a promising application domain, given its prevalence in administrative, financial, and clinical settings and its central role in national data infrastructures and inter-organisational data sharing.

In practical application settings, synthetic data generation typically follows a centralised workflow: relevant data sources are aggregated on a single server, linked via primary and foreign keys, preprocessed (e.g., cleaning and imputation), and used to train a large generative model. The resulting synthetic data are then evaluated to ensure fidelity, utility, and privacy compliance (Yale et al., 2020; Shi et al., 2022; Schneider-Kamp et al., 2024). Some workflows establish formal differential privacy (DP) (Dwork et al., 2006) guarantees at the model level, but in cases where only a finite synthetic dataset is shared (data-centric privacy), DP may indeed be insufficient in quantifying the finite identifiability risk of the finite original data, while introducing an unacceptable amount of noise in the training (Yoon et al., 2020).

An important variant for the centralised paradigm is in the federated or distributed settings, where records and/or features cannot be pooled due to legal, organisational, or technical constraints. In such scenarios, generating high-quality synthetic data remains challenging, and existing approaches for horizontally or vertically federated generation (Fang et al., 2022; Duan et al., 2023; Yuan et al., 2024) generally achieve performance comparable to, but rarely exceeding, their centralised counterparts.

The motivation for disjoint generation was initially prompted by a practical constraint: data supplied in disparate vertical slices with only partial overlap between entries. However, we soon found that partitioning could be beneficial even when working outside the confines of necessity. Computational speed-up is, of course, a trivial and well-known benefit that has been explored in related topics, such as for differential privacy mechanisms (see Hardt et al. (2012)), but other advantages, like access to mixed-model generation and empirical privacy gains, remain unexplored and undocumented. Conditional generation is perhaps the closest related concept, and conditioning based on chunks of already generated variables has been attempted previously in studies with Bayesian network models and auto-regressive generative models (Deeva et al., 2020; Tiwald et al., 2025). Both examples show a limited but promising palette of results with good efficiency and performance compared to their baselines.

Disjoint generation reaches beyond the conditioning setting, which introduces a significant structural challenge with the loss of row-level alignment. Namely, while there remains a relationship in the training data between rows of disjoint tables, synthetic samples will generally not belong together with those modelled based on another slice of the dataset. While this task shares characteristics with record linkage (Smith, 2019) and federated entity matching (Lee et al., 2018), those fields typically rely on some shared information to facilitate probabilistic or rule-based joining. In the truly disjoint setting, where no such overlap ex-

---

[1]The Codebase, experiments, and results are made available at:
https://github.com/notna07/disjoint-synthetic-data-generation

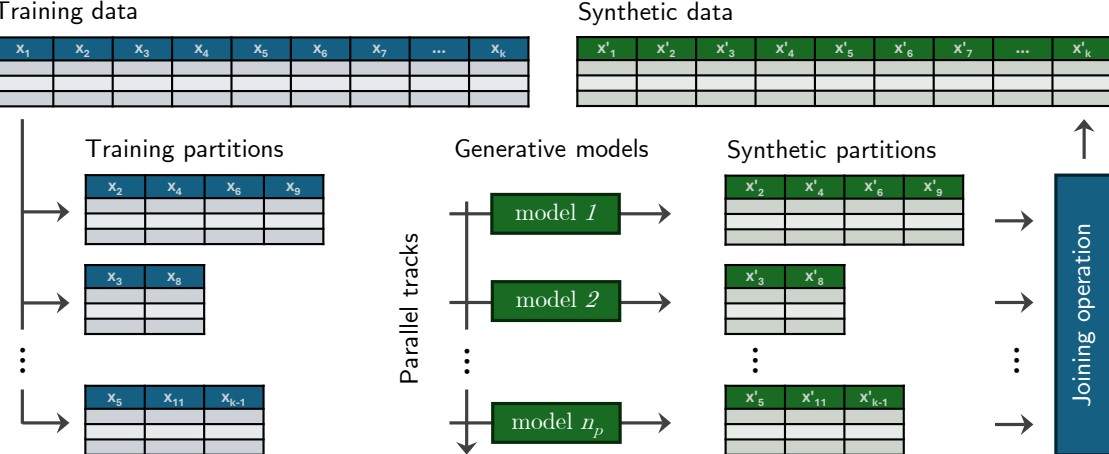

Figure 1: Disjoint generation conceptual overview. The figure shows how our approach splits training data into a number of subsets with similar or different sizes and with no shared variables. Generative models can then be applied in parallel, and the resulting synthetic datasets are then joined using some joining operation into a final output.

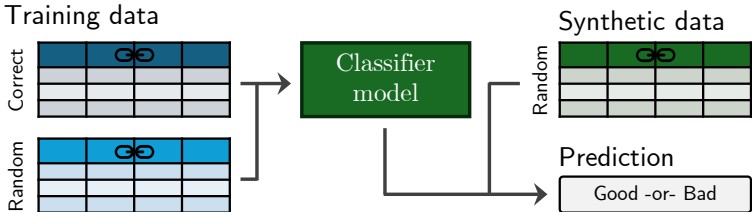

Figure 2: Joining validator training and inference. The validator model is trained in a supervised setting using both correctly and incorrectly joined real data. At inference time, synthetic data is randomly joined, and the validator model assigns a critic score "$z_i$" to each row $q_i$.

ists, the challenge shifts toward learning and preserving underlying cross-slice dependencies. Since machine learning and similarity-based techniques have been shown to outperform traditional probabilistic methods in related domains (Wilson, 2011; Tripathi et al., 2024), they provide a promising foundation for the reassembly framework we propose here.

## 3 Disjoint generative models

The proposed disjoint generative models framework offers an alternative route to generating tabular synthetic datasets. Instead of using one generative model for the synthesis, DGMs distribute the labour by handling disjoint partitions of variables using different generative models and/or model instances and combining the results post hoc (see Figure 1).

Let the initial dataset $X$ be an $(n \times k)$ matrix consisting of $n$ independent observations in $k$ variables, i.e., $X = \{x_1, x_2, \ldots, x_k\}$, where $x_i$ for $i = 1, 2, \ldots, k$ is the feature vectors. $X$ is allocated into the required number of column-wise partitions $n_p$, using some assignment function $r(\cdot)$ which distributes attributes according to some specifications or by random assignment. Next, each dataset partition $d_p$ is handed to a unique generative model instance $G_p(\cdot)$, which creates a synthetic dataset partition $s_p$. Note that in practice, it may be helpful to oversample $s_p$ such that $n < |s_p|$, to ensure enough valid matches are possible in the joining operation. As mentioned previously, this paper treats the special case when the joining operation is done using a model for validating joins. We define $V(\cdot)$, the validator model, as an object that assigns a score $z$ to a queried join $q$ to be a valid observation. In the case where it is a binary classifier, we train $V$ on $X$ as well

---

**Algorithm 1** Disjoint generative models with joining validator. The dataset $\boldsymbol{X} = \{\boldsymbol{x}_1, \boldsymbol{x}_2, \ldots, \boldsymbol{x}_k\}$ consists of $k$ variables and $n$ records, that are to be distributed into $n_p$ disjoint partition column-sets by assignment function $\mathtt{r}(\cdot)$.

---

**Input:** *Dataset* : $\boldsymbol{X}$, *Generative Models* : $\{\mathtt{G}_p \mid p = 1, 2, \ldots, n_p\}$
**Output:** *Dataset* : $\boldsymbol{S}$
1:   $\boldsymbol{S} \leftarrow \emptyset$
2: **for** $p \leftarrow 1, 2, \ldots, n_p$ **do**
3:      $d_p = \{\boldsymbol{x}_i \in \boldsymbol{X} \mid \mathtt{r}(\boldsymbol{x}_i) = p\}$
4:      $s_p \leftarrow \mathtt{G}_p(d_p)$                      ▷ Generative model $\mathtt{G}_p$ makes
5: **end for**                                  synthetic dataset $s_p$.
6: **while** $|\boldsymbol{S}| \leq |\boldsymbol{X}|$ *and* $s_p \neq \emptyset$ **do**
7:      $\boldsymbol{Q} \leftarrow [s_1|s_2|\ldots|s_{n_p}]$
8:      $\boldsymbol{z} \leftarrow \mathtt{V}(\boldsymbol{Q})$            ▷ Validator $\mathtt{V}(\cdot)$ assigns probabilities $\boldsymbol{z}$ to query points.
9:      $\boldsymbol{S} \leftarrow \boldsymbol{S} \bigcup \{\boldsymbol{q}_i \in \boldsymbol{Q} \mid z_i \geq \theta\}$         ▷ Valid items at threshold $\theta$ are saved.
10:      **for** $p \leftarrow 1, 2, \ldots, n_p$ **do**            ▷ But they are removed from the pool.
11:          $s_p \leftarrow s_p \setminus \{s_{p_i} \in s_p \mid z_i \geq \theta\}$
12:          $s_p \leftarrow \mathtt{shuffle}(s_p)$
13:      **end for**
14: **end while**

---

The generative models ($\mathtt{G}_p$'s) can be any arbitrary combination of generative models that work on the data elements (e.g., columns) they get assigned. Similarly, the validator model $\mathtt{V}(\cdot)$ can be any sort of oracle function that assigns a score to the paired-up data elements.

---

as $\boldsymbol{X}' = [d'_1|d'_2|\ldots|d'_{n_p}]$ where the prime indicates that the partition has been randomly shuffled row-wise independently of the other partitions. Artificial labels are assigned accordingly (i.e., 0 for the random joins $\boldsymbol{X}'$ and 1 for authentic joins $\boldsymbol{X}$) and used to train the validator (see Figure 2).

The final step of using the joining validator involves repeatedly creating a query dataset $\boldsymbol{q} \in \boldsymbol{Q} = [s_1|s_2|\ldots|s_{n_p}]$ and obtaining scores $\mathtt{V}(\boldsymbol{Q}) = \boldsymbol{z}$. These scores are used to remove validated query points (defined by some threshold $\theta$) from the query dataset and add them into the output dataset $\boldsymbol{S}$ before the partitions in $\boldsymbol{Q}$ are re-shuffled row-wise and independently to create new candidate joins from the remaining items. This last process is repeated until termination criteria are met ($\boldsymbol{S}$ is large enough, $\boldsymbol{Q}$ is empty, or maximum iterations are reached). The procedure is formally presented in Algorithm 1.

Evidently, several aspects of the presented framework can be changed, altered and improved, opening up additional possibilities. Our experiments explore two algorithmic variations of the joining procedure: using a validator model (a *random forest classifier* from `scikit-learn`) to validate the joins as described, and a random concatenation baseline that foregoes the validation loop (in the algorithm, replace lines 6-14 with one line "$\boldsymbol{S} \leftarrow [s_1|s_2|\ldots|s_{n_p}]$"). While these methods highlight the efficacy of DGMs, many other joining and/or validation methods could be explored in the future. Some software details of our implementation[2] may be found in Appendix A.

## 4   Experiments

In the following, we present experiments with the *Disjoint Generative Models (DGMs)* framework, applied to a handful of benchmark datasets. We include seven common benchmark datasets with a varying number of records (126–3927) and features (11–34), and with a mix of categorical and numerical features. We also included a single high-dimensional (98 features) dataset for some of the experimentation. The details of the datasets are outlined in Table 1.

To provide an overview of the experiments explored in this paper and in the appendices, we here summarise:

---

[2]The repository linked previously holds the implementation, tutorial, and codebooks to reproduce the experimental results.

Table 1: Datasets used in the experiments. The table overviews the datasets used in the various experiments. They are arranged alphabetically, and the number of records/attributes are shown together with a breakdown of attribute types. The diabetic mellitus dataset is used for higher-dimensional experiments only.

| Dataset identifiers | | | # of records | | # of atts. | |
|---|---|---|---|---|---|---|
| key | Name | Source | Train | Test | Categorical | Numerical |
| al | alzheimer's disease | kaggle | 1719 | 430 | 18 | 15 |
| bc | breast cancer | kaggle | 3219 | 805 | 11 | 5 |
| cc | cervical cancer | UCI | 534 | 134 | 26 | 8 |
| hd | heart disease | UCI | 242 | 61 | 9 | 5 |
| hp | hepatitis | UCI | 1105 | 276 | 17 | 12 |
| kd | kidney disease | UCI | 126 | 32 | 14 | 11 |
| st | stroke | kaggle | 3927 | 982 | 8 | 3 |
| dm | diabetic mellitus | OpenML | 225 | 56 | 93 | 5 |

* The hepatitis dataset had its multilayered label column binarised to case/no-case.

**4.1 Partitioning improves privacy but harms utility.** The first experiment explores the premise of the idea: how using the same generative model, privacy improves while utility decreases with every additional partition. Appendix B contextualises these findings with forms of noise injection.

**4.2 Large dataset and using joining validator.** This experiment shows two results: that the behaviours we observed for the smaller benchmark datasets persist for a higher-dimensional dataset, and that using the joining validator is more conservative on utility than random concatenation.

**4.3 Partitioning improves efficiency of some models.** We demonstrate as a corollary result of the previous two experiments, that for models that scale in the number of variables, partitioning can (obviously) improve training time.

**4.4 It matters which variables end up in which partition.** This experiment shows that the relationships between variables in different partitions are important for both the performance of the joining validator and random concatenation joining.

**4.5 Mixed model generation.** This experiment demonstrates how using a different model for each partition can yield a dataset that balances utility and privacy.

**C.1 Using different back-ends for the joining validator.** This experiment in the appendix explores whether some choices of backend for the joining validator model are better than others. We did not find any overall best model; there were some which stood out on the dataset level, but the random forest model seems as good a choice as any for seeing what DGMs can do.

**C.2 Validator effectiveness, hyperparameters optimisation, and calibration.** This appendix discusses the importance of conducting hyperparameter optimisation and calibration. We demonstrate that under the simplest of conditions, an unoptimised validator model may yet introduce artefacts in the joining process and produce synthetic data with poor statistical fidelity.

**C.3 Static acceptance threshold setting.** This experiment explores how the quality of the sampled dataset is affected based on the threshold of acceptance, which is used in the validator model. The results once again emphasise the importance of proper optimisation and calibration.

**Please note.** While we do indeed employ some models with differential privacy throughout this paper, our evaluation focuses on empirical, dataset-level privacy and re-identification risk, and does not aim to provide formal differential privacy guarantees at the model level. Establishing end-to-end differential privacy for DGMs, including all constituent submodels and validation components, remains an open problem and should be explored in future works.

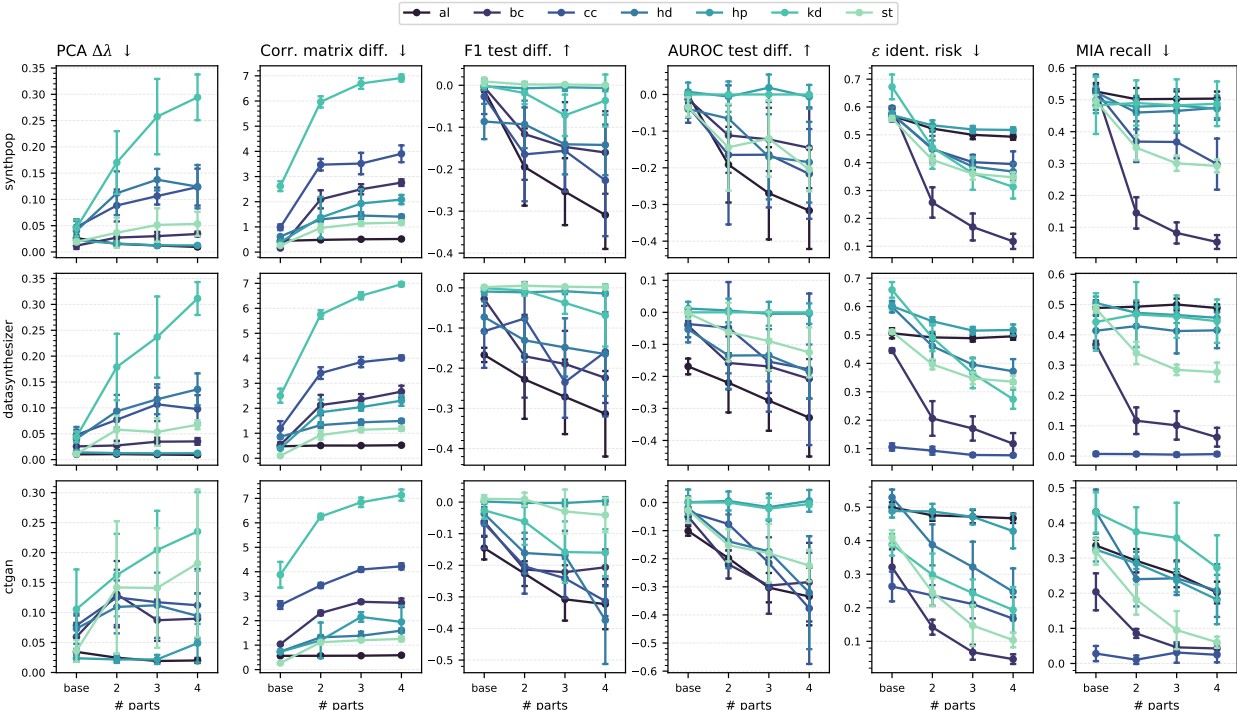

Figure 3: Evaluation metrics vs. number of partitions. The figure shows the result of 10 repeated experiments using same-model DGMs with an increasing number of equal-sized random partitions of the column-sets. Error bars denote standard deviation. While fluctuations and single datasets deviate from the group behaviour, utility (PCA eigenvalue, correlation matrix, hold-out F1, and AUROC difference) generally worsens, while privacy ($\varepsilon$-risk and MIA recall) improves as the number of partitions increases. Values closer to zero are better.

To evaluate synthetic tabular data in the experiments below, we use the SynthEval evaluation framework (Lautrup et al., 2024b), with a broad selection of recognised metrics for utility and privacy (Hernandez et al., 2022; Dankar et al., 2022; Lautrup et al., 2024a; Hyrup et al., 2025). For measuring utility, we use PCA eigenvalue- and eigenvector angle difference (Rajabinasab et al., 2025), Hellinger distance, correlation matrix difference, AUROC difference, and accuracy difference for training and holdout set. We estimate empirical privacy-preserving qualities using $\varepsilon$-identifiability risk[3] (Yoon et al., 2020), median of the distance to closest record (DCR), and precision and recall of a "worst case" membership inference attack (MIA) model. A brief explanation of each metric is provided in Appendix D.

To account for experimental randomness, we left the random seed unlocked and conducted repeated experiments to ensure robust measurements. The error bars presented denote the unscaled unit of variation, standard deviation or standard error, as stated in the figure caption.

## 4.1 Partitioning improves privacy but harms utility

This first series of experiments demonstrate the cornerstone idea of the DGMs framework, namely, that privacy of synthetic data improves to the detriment of utility when partitioning the training data column-wise. We show the effect under the simplest conditions with a growing number of equal-sized random partitions processed by different instances of the same generative model and then concatenated randomly back together. This can indeed be viewed as a form of noise introduced in the generative process by fragmenting information; in Appendix B we extend the present experiment to other types of noise injection. The results of partitioning are presented in Figure 3, for a selection of datasets using three different types of

---

[3]Not to be confused with $\epsilon$-differential privacy.

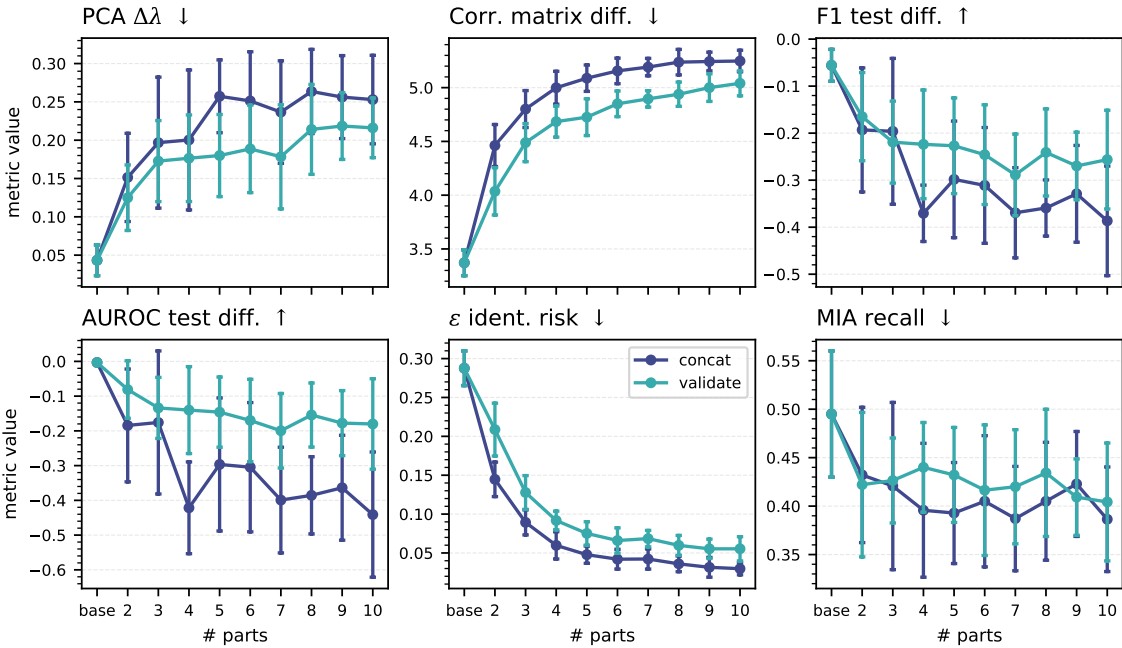

Figure 4: Joining operations by metric vs. number of partitions. The figure shows the results of concatenation on the diabetic mellitus high-dimensional dataset generated with disjoint DataSynthesizer models to show that the expected behaviour persists for more dimensions and partitions than presented in the previous figure. Additionally, the results from using a random forest classification model for joining validator are shown for the same dataset, illustrating that deliberately choosing joins that look more authentic can reduce the utility loss at the cost of some of the privacy gains. Error bars denote standard deviation from across 20 repeated experiments.

generative models: The `synthpop` (Nowok et al., 2016) sequential Classification and Regression Tree (CART) model, the DataSynthesizer (Ping et al., 2017) Bayesian Network (BN) model, and a Generative Adversarial Network (GAN) model CTGAN (Xu et al., 2019), representing typical choices for three species of tabular generative models. The experiments generally agree with our hypothesis, although datasets occasionally show only a weak signal on the level of individual models or metrics. This experiment shows that there is potential in DGMs if we can, in some way, control the loss in utility while keeping privacy gains.

## 4.2 Large dataset and using joining validator

Next, we apply the validation scheme described above, but for brevity, we focus on the DataSynthesizer BN model only on a single high-dimensional dataset ("dm" in Table 1). The results shown in Figure 4 show that the effect from before appears to persist on this larger dataset with more partitions than what was possible on the smaller datasets. Additionally, the result of applying the validation scheme (recall that we use a *random forest classifier*, cf. C.1) to assess the "realness" of randomly concatenated query joins seems to improve utility, while also negatively affecting privacy. The error bars denote the standard deviation from 20 repeated experiments and show that there is some variability in the performance, subject to which attributes are assigned to which partition. We note that good and bad final results *can* happen by chance for both joining methods, but generally, results based on validated joins are preferable on all metrics for any number of partitions.

## 4.3 Partitioning improves efficiency of some models

Another effect of partitioning that was particularly apparent for the high-dimensional dataset was that partitioning can improve the efficiency of some generative methods. This is, of course, not a surprising

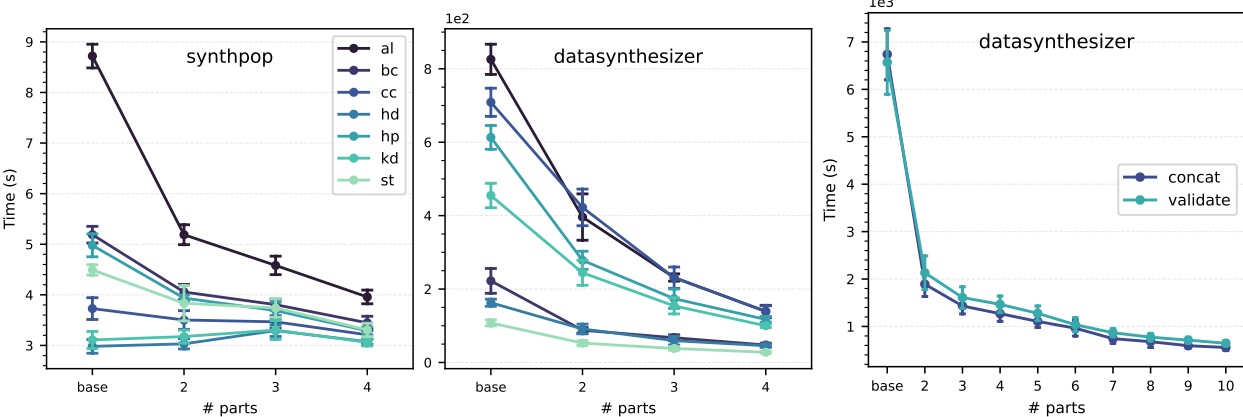

Figure 5: Effect of disjoint generation on running time. The plots show the process time measured for each experiment (not conducted in an isolated environment) with the `synthpop` and DataSynthesizer models used for all partitions. First two frames are from the first experiment with only concatenation of the partitions, and the last frame is for the second part with only the DataSynthesizer model on the high-dimensional dataset.

result; partitioning is known to increase algorithmic efficiency. However, not all generative models are equally efficient; for example, the Bayesian network model applied here is not the obvious first choice for a high-dimensional dataset. While not exactly combinatoric in the attribute number due to efficient heuristics, graph-based models are sometimes avoided for larger projects due to their poor scaling. By applying the DGMs framework, it is perhaps obvious that $O\left(n_p \left(\frac{k}{n_p}\right)^c\right) \leq O(k^c)$ for arbitrary positive $c$, and also $O\left(n_p \left(\frac{k}{n_p}!\right)\right) \leq O(k!)$; in other words, models that scale poorly can be made viable by using partitioning and disjoint generation.

We observe this benefit for the `synthpop` sequential CART model (Nowok et al., 2016) (which was fast already) and the DataSynthesizer BN model (which benefited significantly) in our experiments (see Figure 5). It should be noted, that none of these timing measurements were conducted rigorously in an isolated environment, and while they follow the postulated behaviour, they are nevertheless of a more anecdotal and/or corollary nature. The results from the GAN models used in this paper, CTGAN and DPGAN (Xie et al., 2018), are not presented for this reason, since we were unable to measure them for different partition numbers at a consistent load due to substantial overhead from training multiple big neural networks in parallel.

## 4.4 It matters which variables end up in which partition

For the next part, we abandon the randomly selected partitions and consider whether we can be a bit more deliberate in grouping the variables together. Above, we already recognised that for variables randomly assigned to partitions, there is a chance that concatenation may work equally well or better than validation. This can be caused by multiple factors, such as the fit quality of the validator model (treated in Appendix C.2), or more importantly, due to the strength (or relative weakness) of associations between features assigned to *different* partitions. As trivial as it might seem, if partitions have no shared information, correlation, or other patterns, then concatenation will be more successful, as there are no inter-partition relationships to misrepresent. Using the joining validation, on the other hand, will be difficult because there are no patterns to learn between the partitions; as a result, the model may be susceptible to overfitting and introducing spurious biases, which makes the quality of the generated data worse. Conversely, ensuring that partitions have more of a co-dependency makes the job of the validator model easier while worsening the results from using concatenation as a joining strategy.

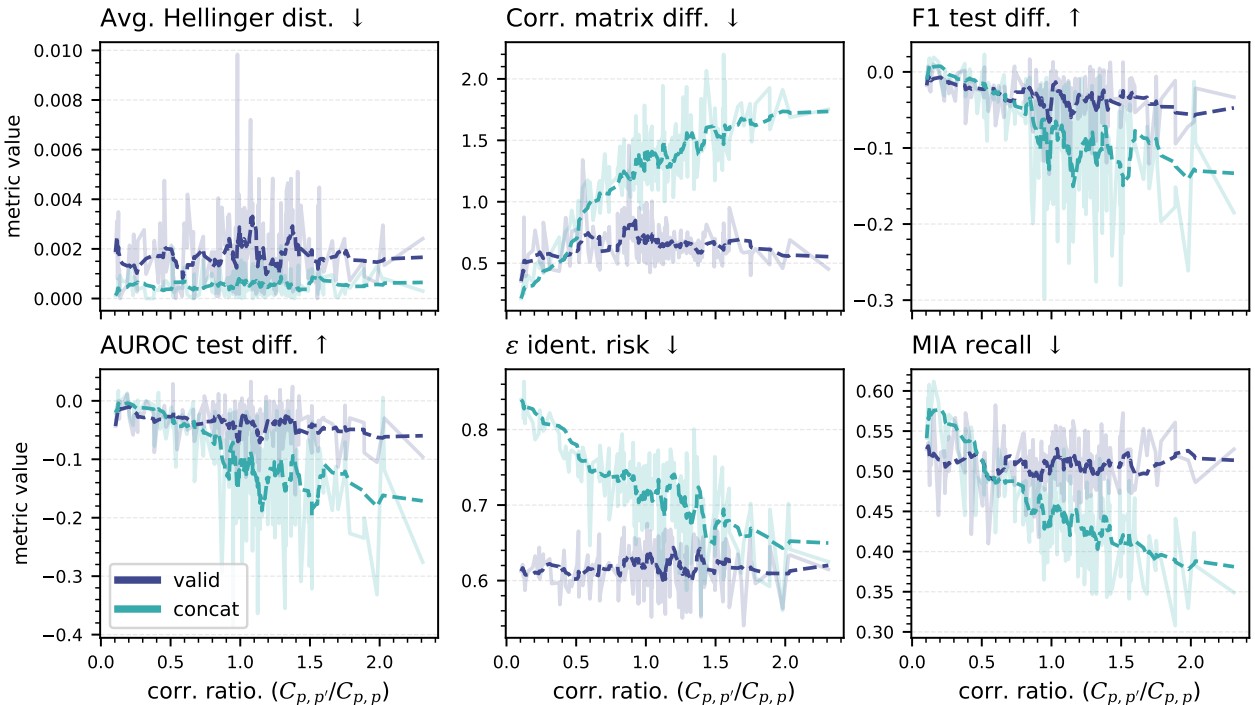

Figure 6: Effect of inter-partition correlation on various metrics. The figure shows results and moving averages from DGMs using two different joining schemes on random dummy data sampled with varying levels of correlation between two partitions. The number on the x-axis is the ratio of exterior correlations to interior correlations measured by the Frobenius norm. The dashed lines are moving averages (window size 10) on a background of the actual measurements.

In order to have more control in studying this effect, we used dummy data, where the relative strength of the partition cross-correlations could be adjusted. In Figure 6, we show the effect of applying DGMs with the `synthpop` model to such data sampled from 210 separate random matrices sorted by progressively stronger correlation. For the most part, the observed tendencies align with our assessment; when the exterior correlations are much weaker than the interior correlations, concatenation is the superior method, whereas when the importance of the cross-correlations grows, joining with the validation model becomes more feasible and provides better utility. Concatenation consistently performs better on the Hellinger distance metric because it does not affect the marginal distributions of the variables in the joining process. Validation, on the other hand, arguably introduces a slight bias in the candidates accepted, which makes the marginal distributions less similar to the originals (more on this in Appendix C.2). Privacy metrics ($\varepsilon$-identifiability risk and MIA recall) are worse for concatenation joining early on and improve by $\approx 0.2$ throughout the experiment. This means that it becomes more difficult to hit a real data point by accident when cross-correlations are present. The validator model does not tend remarkably in any direction as the cross-correlations increase.

Based on these findings, it would arguably be practical to ensure a high degree of dependency is shared between the partitions if using validated joins. Alternatively, if using concatenation joining, as little information as possible should be shared among the partitions in order to ensure seamless joining.

## 4.5 Mixed model generation

One of the primary motivations of the disjoint generation approach is the ability to use any combination of generative models to create synthetic data. This allows for the selection of the best models for each subproblem (e.g., data types, sensitive variables, or domain challenges) in practical application without

Table 2: Hepatitis dataset, high correlation partitioning. The hepatitis dataset (Kamal et al., 2019) was partitioned according to the high correlation partitioning scheme described in the text. Asterisk marks attributes that were made discrete.

|  | Attributes assigned to the partition. |
|---|---|
| part1 | *RNA EOT, ALT24(24 weeks), Diarrhea, BMI(Body Mass Index), Age, Headache, *Plat(Platelet), Fatigue & generalized bone ache, Nausea/Vomiting, Gender, ALT36(36 weeks), Fever, Epigastric pain, AST1(1 week), *RNA Base, b_class (binarised original multilayered label). |
| part2 | *RNA EF(Elongation Factor), ALT48(48 weeks), HGB(Hemoglobin), *RNA 12, ALT1(1 week), ALT12(12 weeks), ALT4(4 weeks), *RNA 4, Baseline Histological Grading, Jaundice, *RBC(Red Blood Cells), *WBC(White Blood Cells). |

having to compromise by choosing a single model overall. Models that enable differential privacy, for example, tend to sacrifice utility in favour of the theoretical privacy guarantee, whereas high-utility models frequently make the opposite choice (Yoon et al., 2020). In the following, we consider the Hepatitis dataset (Table 1, `hp`) as a case and show how partitioning based on correlation can be used to achieve superior empirical privacy at a more acceptable utility trade-off. The attributes are categorised[4] as shown in Table 2. This partitioning was created by iteratively finding the largest element in a correlation matrix, assigning the constituent pair of attributes to separate partitions, and then removing the corresponding row and column from the correlation matrix, repeating this process until the matrix was emptied. In the present case, this gives us a ratio of exterior to interior correlations of 1.62.

For our experiments, we choose to focus on four generative models, namely, `synthpop` and DataSynthesizer from before, alongside DPGAN (Xie et al., 2018) and TabDiff (Shi et al., 2025); this gives us a total of 12 different mixed-model combinations we can check for the present partitioning. Synthpop and TabDiff are high-utility choices, representing a classical machine learning method and a state-of-the-art (cf., Jiang et al. (2026)) deep learning architecture respectively. DataSynthesizer and DPGAN are options that have differential privacy guarantee enabled (different strengths on the default parameter settings). For baselines, first, we apply all models non-disjointly to the full column-set; second, we add examples of both parts generated by different instances of the same generative model, and third, as an additional comparison, we provide full column-set generation results for models: CTGAN and TVAE (Xu et al., 2019), ARF (Watson et al., 2023), ADS-GAN (Yoon et al., 2020), and TabDDPM (Kotelnikov et al., 2023). Here, we present results for validated joins; results for concatenation are available in the code supplement.

The results are presented in detail in Table 3. The baselines are mostly as expected; the `synthpop` model performs well on the statistical metrics, acceptably on the machine learning metrics, and poorly on privacy; TabDiff reaches a similar performance while for the DPGAN model, it performs worse on the statistical utility and well on privacy. DataSynthesizer finds somewhat of a middle ground but fails to achieve sufficiently low privacy (gauged by the often-mentioned 9% identification risk that various agencies suggest to be acceptable for public release; see European Medicines Agency (2018); Health Canada (2019)) while degrading utility noticeably. The single-model DGMs arrange themselves similarly, with some minor differences. The additional baseline models arrange themselves consistently for "high utility" models with results in the same range as the non-differentially private `synthpop` model. Only TabDDPM sets itself apart from the group with weaker utility performance and intermediate privacy results[5].

---

[4]To get some more nuance with the correlated partitions, we discretised some of the values (marked in Table 2 with asterisks) based on the specification in the dataset supplement file (Kamal et al., 2019). We chose not to discretise all of the numerical attributes but only those with extreme values (ranging in the thousands and millions).

[5]We know that TabDDPM can usually obtain better results than what we see on this dataset (cf. Kotelnikov et al. (2023); Hansen et al. (2023); Jiang et al. (2026), and Figure 8). We did optimise the hyperparameters; these are the best we found.

Table 3: Multi-axis benchmark of generative models modelling hepatitis dataset. The parentheses hold the errors of the last significant figure, measured across 10x repeated experiments. **Bold** values mark the best result in a column within each section; *italics* denote statistically indistinguishable results (95% CI). Most metrics are better when lower, except for AUROC, F1 train, and F1 test. diff., and mDCR. Negative/positive values on the prediction metrics indicate whether the synthetic data are worse/better than the real data for downstream classification tasks. The row highlighted in blue is the best average model.

| | Utility metrics | | | | | | | Privacy metrics | | | | |
|---|---|---|---|---|---|---|---|---|---|---|---|---|
| Model | $\Delta\lambda$ | $\Delta\alpha$ | H-dist. | Corr. diff. | AUROC diff. | F1 train diff. | F1 test diff. | $\varepsilon$ risk | $\varepsilon$ loss | mDCR | MIA re | MIA pr |
| **Non-Disjoint Model Baselines** | | | | | | | | | | | | |
| sp | *0.022*(2) | **0.42**(6) | **0.0058**(4) | *0.397*(14) | *-0.016*(9) | -0.083(2) | *-0.007*(2) | 0.561(3) | 0.361(3) | 0.95(2) | 0.527(13) | 0.528(8) |
| td | **0.018**(2) | *0.58*(6) | 0.0094(5) | **0.379**(10) | *-0.004*(11) | **-0.068**(3) | *-0.005*(4) | 0.512(8) | 0.308(8) | 0.904(15) | 0.500(11) | 0.520(7) |
| ds | 0.077(4) | 0.90(3) | 0.102(5) | 2.88(9) | *0.003*(12) | -0.084(6) | **0.013**(9) | 0.278(7) | 0.125(5) | 1.6(2) | 0.022(4) | 0.49(7) |
| dp | 0.28(3) | 0.85(6) | 0.273(4) | 2.68(2) | **0.014**(11) | -0.304(13) | -0.215(13) | **0.027**(8) | **0.010**(4) | **2.00**(3) | **0.0003**(3) | **0.02**(2) |
| **Extra Non-Disjoint Model Baselines** | | | | | | | | | | | | |
| arf | *0.0238*(10) | *0.59*(7) | **0.0065**(2) | 2.02(2) | *0.013*(8) | *-0.076*(5) | *0.003*(3) | 0.527(2) | 0.328(2) | 0.997(3) | 0.410(7) | *0.519*(6) |
| tvae | **0.021**(2) | **0.57**(8) | 0.0179(9) | 1.62(3) | -0.016(5) | *-0.073*(2) | -0.003(3) | 0.568(5) | 0.360(7) | 0.942(4) | **0.162**(5) | **0.513**(10) |
| ct | *0.024*(2) | *0.67*(8) | 0.0140(8) | **0.75**(2) | **0.025**(9) | *-0.077*(4) | *0.003*(3) | 0.485(5) | 0.286(6) | 0.988(5) | 0.330(9) | *0.519*(8) |
| ad | *0.023*(2) | **0.57**(6) | 0.0117(6) | *0.76*(2) | *0.013*(7) | **-0.068**(3) | **0.008**(3) | 0.497(7) | 0.296(6) | 0.992(3) | 0.35(2) | *0.515*(10) |
| ddpm | 0.103(5) | *0.77*(5) | 0.027(2) | 1.88(11) | -0.032(8) | -0.090(4) | -0.012(2) | **0.251**(11) | **0.115**(7) | **1.16**(2) | *0.167*(13) | *0.542*(8) |
| **Disjoint Generative Models, Single Model, Joining Validator** | | | | | | | | | | | | |
| (sp,sp) | **0.031**(2) | **0.56**(5) | 0.0169(6) | **0.59**(2) | **0.008**(12) | *-0.076*(2) | **0.001**(3) | 0.535(4) | 0.336(4) | 0.950(2) | 0.471(8) | 0.523(5) |
| (td,td) | *0.036*(2) | *0.58*(2) | **0.0087**(5) | 0.90(2) | *0.007*(9) | **-0.074**(3) | *-0.007*(2) | 0.466(3) | 0.268(3) | 1.007(2) | 0.441(10) | 0.522(7) |
| (ds,ds) | 0.068(3) | 0.81(4) | 0.097(4) | 2.44(10) | *0.003*(8) | *-0.088(13)* | **0.001**(20) | 0.317(9) | 0.154(8) | 1.49(2) | 0.031(3) | 0.60(3) |
| (dp,dp) | 0.20(2) | 0.85(6) | 0.284(6) | 2.56(3) | *0.001*(11) | -0.27(2) | -0.19(2) | **0.048**(7) | **0.017**(4) | **1.93**(4) | **0.0006**(4) | **0.04**(3) |
| **Disjoint Generative Models, Mixed Models, Joining Validator** | | | | | | | | | | | | |
| (sp,td) | **0.030**(2) | *0.62*(4) | 0.0103(4) | 0.92(2) | 0.008(7) | *-0.064*(4) | *-0.0079*(14) | 0.482(5) | 0.283(6) | 0.9992(13) | 0.447(7) | 0.520(4) |
| (sp,ds) | 0.060(2) | **0.54**(4) | 0.058(2) | 2.08(8) | *0.018*(12) | *-0.073*(4) | *0.008*(4) | 0.363(7) | 0.193(6) | 1.32(3) | 0.066(10) | 0.57(2) |
| (sp,dp) | 0.30(2) | *0.55*(5) | 0.139(5) | 2.22(5) | *0.016*(10) | *-0.082*(9) | *0.002*(4) | *0.13*(2) | *0.071*(14) | *1.70*(2) | **0.0006**(5) | **0.04**(3) |
| (ds,sp) | 0.041(3) | 0.81(5) | 0.048(2) | 1.33(13) | **0.045**(11) | *-0.10*(2) | *0.007*(13) | 0.474(7) | 0.280(6) | 1.075(13) | 0.16(2) | 0.48(2) |
| (ds,td) | 0.050(3) | 0.87(3) | 0.053(6) | 1.6(2) | -0.008(10) | **-0.061**(13) | **0.010**(12) | 0.419(7) | 0.231(6) | 1.113(10) | 0.16(2) | 0.477(14) |
| (ds,dp) | 0.27(2) | 0.76(7) | 0.188(9) | 2.59(6) | *0.034*(11) | *-0.11*(2) | *-0.001*(16) | *0.15*(3) | *0.07*(2) | **1.75**(4) | *0.006*(4) | *0.23*(9) |
| (td,sp) | *0.032*(2) | *0.61*(3) | **0.0079**(6) | **0.79**(2) | *0.010*(14) | *-0.077*(3) | *-0.007*(2) | 0.515(2) | 0.311(2) | 1.000(2) | 0.468(9) | 0.515(6) |
| (td,ds) | 0.056(2) | *0.58*(2) | 0.066(2) | 2.48(8) | 0.001(10) | *-0.075*(2) | *-0.003*(2) | 0.218(5) | 0.094(4) | 1.46(3) | 0.024(4) | 0.51(7) |
| (td,dp) | 0.278(13) | *0.57*(6) | 0.152(4) | 2.40(5) | *-0.01*(2) | *-0.070*(3) | *0.008*(4) | *0.12*(2) | *0.060*(9) | **1.75**(2) | *0.0035*(12) | *0.20*(7) |
| (dp,sp) | 0.27(2) | *0.66*(9) | 0.167(6) | 2.32(8) | *0.001*(14) | -0.23(4) | -0.14(4) | *0.14*(3) | *0.07*(2) | *1.64*(6) | *0.0038*(13) | *0.17*(7) |
| (dp,td) | 0.247(11) | 0.78(4) | 0.157(8) | 1.34(14) | *0.022*(8) | -0.27(3) | -0.18(4) | 0.189(13) | 0.102(9) | 1.24(3) | 0.015(3) | 0.43(6) |
| (dp,ds) | 0.24(2) | 0.87(3) | 0.223(8) | 2.38(12) | *0.011*(11) | -0.24(4) | -0.15(4) | **0.114**(11) | **0.05**(5) | 1.61(4) | 0.0061(15) | 0.28(8) |

Model shorthands: sp - synthpop, td - TabDiff, ds - DataSynthesizer, dp - DPGAN, ct - CTGAN, ad - ADS-GAN, ddpm - TabDDPM.

The mixed model results are presented in the last section of Table 3. Five of these combinations are particularly noteworthy for privacy, improving significantly over the synthpop, TabDiff, and DataSynthesizer results in the baselines. The combinations with DPGAN for modelling "part1" remain rather unimpressive for utility; conversely, the synthpop-DPGAN, TabDiff-DPGAN, and DataSynthesizer-DPGAN DGMs only barely miss the mark for acceptable privacy whilst posing much preferable utility in comparison. Out of the three, the synthpop-DPGAN and TabDiff-DPGAN DGMs place slightly better on key metrics (9/12 top results) and may be good choices for overall balanced models.

It is curious to observe how the same two generative models can give different results when put in charge of different parts of the data. From what we can tell, it matters what generative model is put in charge of the partition with the label variable; the stronger privacy enforced by DPGAN significantly affects ei-

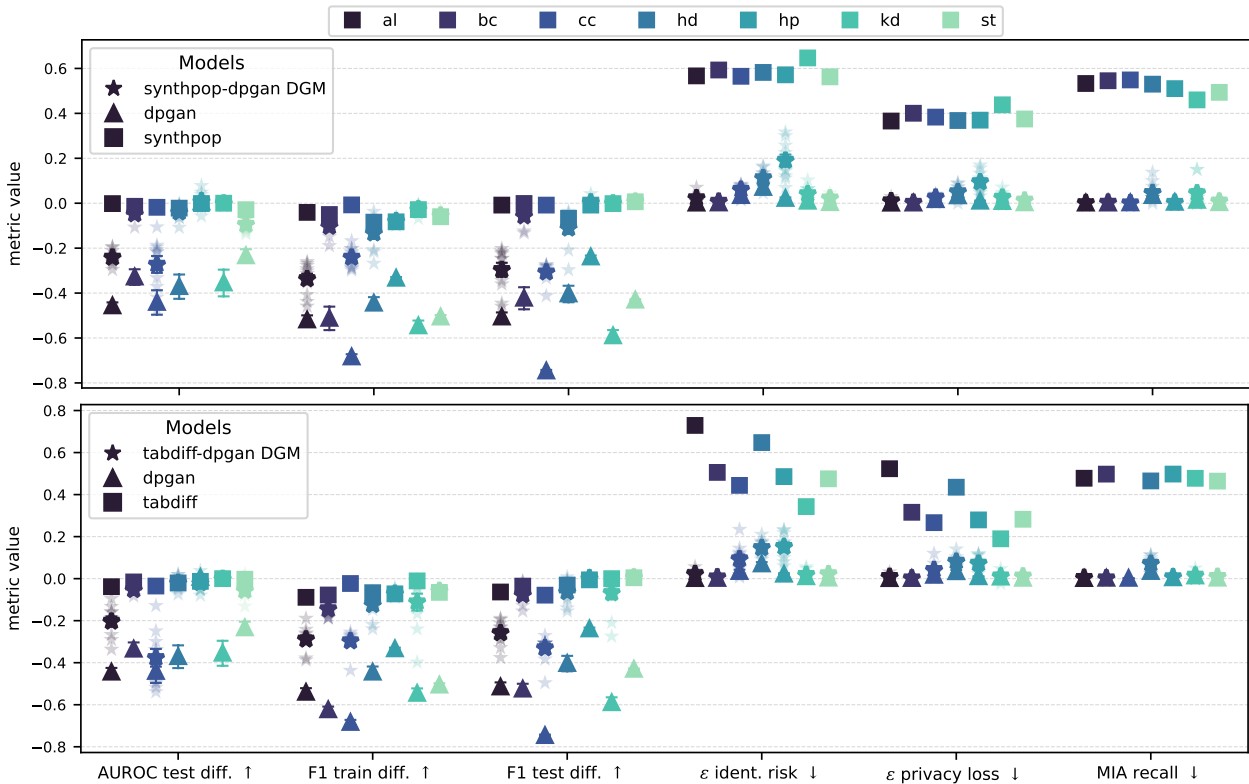

Figure 7: Results for mixed-model generation. Top: The result from 10x repeated synthesis of the benchmark datasets (Table 1), using synthpop, DPGAN, and synthpop-DPGAN DGM (partitioned using high-exterior correlation scheme). As seen in the figure, the DGM always places a better position than the DPGAN model on utility and is comparable to it on privacy. Bottom: Same experimental setup and presentation, for TabDiff-DPGAN DGM. Error bars denote standard error. Faint star icons are individual measurements of the DGMs' results. The arrows next to the metric names indicate the positive direction.

ther elements of the joining process or the evaluation itself. Moreover, some models are better suited for generating certain data types. For example, GAN models can underperform when many variables are discrete, multimodal, or imbalanced (Xu et al., 2019): In Table 2, "part1" happens to contain more categorical attributes than numericals and vice versa for "part2". In future studies, the particular effects of which variables and generative models are assigned to which partitions should be further examined. Initially, we explored splitting on categorical and numerical attributes, and while it was effective, exterior correlations with this partitioning were generally on the lower end for these typical benchmark datasets, resulting in underconfidence for the validator models.

Before discussing some limitations, it is beneficial to demonstrate that the results presented in the above case study also hold for the other datasets. We run the synthpop-DPGAN and TabDiff-DPGAN DGMs on the `al, bc, cc, hd, kd, st` datasets from earlier and show the results in Figure 7 together with the baseline measurements of the respective models for 10 times repeated experiments. In the figure, we omit the statistical metrics to present a focused display of the ML metrics and privacy dimensions; the full results can be found in the code supplement. Again, there is some dataset variability; the alzheimer's disease dataset and cervical cancer are the worst examples, but the DGM utility results still place better than the DPGAN on average while providing much better privacy than for the high-utility models across the board. We show the individual measurements from the DGMs as faint points to illustrate the variability. In Figure 8 we overlay the synthpop-DPGAN DGM result with the additional baseline models for further comparison. Again the DGM example demonstrate competitive performance with only a few exceptions.

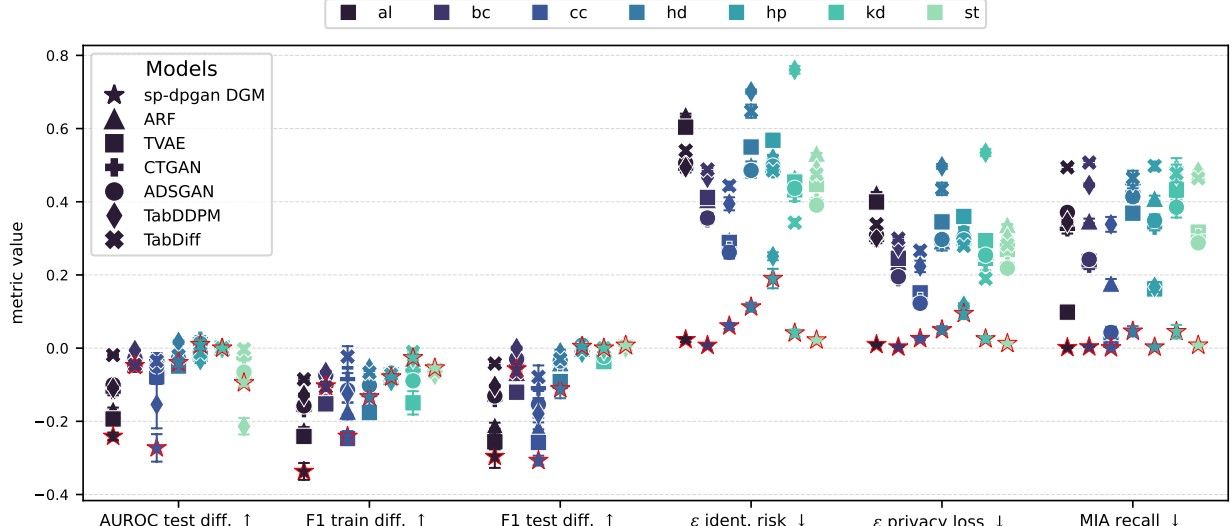

Figure 8: Baseline comparison with mixed-model generation. This figure shows extra baseline model results: the synthpop-DPGAN DGM result from before (Figure 7) is added on top with a red outline. With some exceptions (particularly the `al` and `cc` datasets), the mixed model places well within the group of baseline models on the utility metrics, but is considerably ahead on the privacy metrics. Error bars denote standard error from 10x repeated experiments. The arrows next to the metric names indicate the positive direction.

In summary, indications are that mixed model DGMs can achieve a better trade-off between utility and privacy than we get from optimising a single model towards privacy. There are, however, notable concerns and limitations to keep in mind:

1. While using the DPGAN and DataSynthesizer models may be thought to provide a differential privacy guarantee, it is important to remember that other parts of the DGM are not differentially private; furthermore, the current joining validator is exposed to real data samples and thus **voids any theoretical privacy guarantees** of the constituent models.

2. While the epsilon identifiability metric and membership inference recall are affected significantly and for some datasets are brought down below the 9% identification risk, **an adversary knowing how the data have been partitioned can compromise the privacy of the data easily**.

3. Training the validator is an exercise in calibration and optimisation, depending on various factors; one model may benefit privacy or utility more or **introduce a selection bias or unfairness** in what kind of items are accepted. Judging from how marginal metrics are affected, it is clear that there is an impact to some properties of the data following the joining procedure; see Appendix C.2.

## 5 Discussion

In this paper, we demonstrate the potential of partitioning in the context of tabular synthetic data generation. We proposed *Disjoint Generative Models (DGMs)*, a framework for splitting data column-wise, training separate instances on generative models on the partitions and joining post hoc. Using this approach with naïve choices for how to partition and join, as we do in our experiments, appears to offer a benefit towards striking a balance of utility and privacy in synthetic data generation, which is worth further consideration.

Our experiments show how increasing levels of partitioning and random reassembly gradually benefit heuristic privacy while negatively affecting utility. We show how using a simple joining validator model to moderate the reassembly can remedy some utility loss, and we see how the overall methodology can significantly reduce

computational overhead with certain model types like Bayesian networks. With mixed model generation, we saw how putting different models in charge of different partitions could harness the strengths of each without having to compromise by choosing a single model overall. Particularly, using high-privacy models such as DPGAN together with high-utility models such as `synthpop` CART or TabDiff allowed the creation of high-utility synthetic data that conforms to current standards for distance-based identification risk.

Our experiments only scratch the surface of possible research directions, and there is certainly much to be explored in future works; obstacles to overcome, experimental boundaries to push, and design choices to master. Arguably, alternative methods for joining the data should be explored, such as Bayesian methods, expectation maximisation, hashing functions, similarity learning, or ordered weighting averaging operators (consider, e.g., Reventós (2004); Lee et al. (2018); Smith (2019)).

Some additional topics and extensions that could be useful to explore are listed below:

- Replacing the generative model library with simple attribute samplers could allow DGMs to be used recursively.

- Evidently, some important statistical and/or fairness properties are negatively impacted by disjoint generation (see Appendix C.2). However, investigating whether fairness can be controlled or augmented by disentangling the protected attributes from the rest could be productive.

- To enhance cross-partition correlation with the class feature, adding the class to every partition could be investigated as either an inherent part of each partition or as a conditioning vector. By extension, conditioning may help the validator model to prevent class imbalances.

- Considering that the number of partitions affects the output quality and stability of DGM, an investigation of the dichotomy between partition size and number of partitions could shed light on finding the right balance.

- Exploring techniques for assigning features and models to partitions systematically could optimise the effectiveness of the framework.

- Establishing end-to-end privacy, by using differentially private constituent models, validation process, and covariance estimation (Amin et al., 2019), while accounting for the privacy budget across the full framework, could widen the appeal of disjoint generation in more regulated settings and for continual release scenarios.

Finally, this study focused only on tabular data, where several established metrics allow for measuring quality. However, this study also paves the way for future investigations of creating fully fledged multi-modal synthetic records consisting of not only categorical and numerical attributes but also images, text, and time-series data.

This paper has demonstrated the viability of the disjoint generative models framework, suggesting that this research direction could prove useful.

**Broader Impact Statement**

Current literature on synthetic data generation identifies a gap, or perhaps a disconnect, between ongoing efforts for high utility and privacy in synthetic data, and the proposed disjoint generative models framework offers a quite direct method for marrying the breakthroughs in both areas through mixed-model generation. Moreover, joining the efforts of specialised models and performances could also unlock a new frontier for multi-modal learning (which was not explored in this work), as state-of-the-art, text, image, tabular, and time-series models may be used in tandem with a multimodal classification model to achieve multi-modal data generation.

However, as we acknowledge, the privacy we treat here is empirical rather than formal and is therefore limited to the synthetic datasets themselves rather than the models that produce them. In other words, the framework in its current state does not provide end-to-end privacy guarantees, and the reported reductions in identification risk may not hold in realistic adversarial scenarios.

## Acknowledgments

This study was funded by Innovation Fund Denmark in the project "PREPARE: Personalized Risk Estimation and Prevention of Cardiovascular Disease".

## Materials Availability

We make the codebase, experiments, and results available as a supplement. The provided material also contains tutorials to reproduce the experimental results. The repository can be found at:

`https://github.com/notna07/disjoint-synthetic-data-generation`.

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

# A    Implementation details

This appendix contains details on the implementation of the disjoint generation framework that we made and use for the experiments[6]. Some practical choices made in the implementation are left out of the main text since they are merely *software* solutions, and are not necessary knowledge to appreciate the effect of partitioning. The effect of disjoint generation has been verified through multiple independent implementations, and the necessary elements to reproduce the main findings are distilled in the formal Algorithm 1.

## A.1    Python library

For investigating the potential of DGMs, and exploring various aspects, we implemented a basic library for disjoint synthetic data generation that interfaces various generative model frameworks, namely, Synthcity (Qian et al., 2023), DataSynthesizer (Ping et al., 2017), TabDiff (Shi et al., 2025), and synthpop (Nowok et al., 2016), with different joining strategies and allows subsets of columns to be easily specified or selected randomly.

In our experiments, we introduce two algorithmic variants: randomly concatenating the synthetic outputs and a simple approach where a validator model assigns a score to the candidate joins. The implementation allows for other strategies to be added as well and for the back-end of the validator module to be set to any classification module with the appropriate methods. We use a random forest classifier from `scikit-learn` by default and leverage the full training dataset (see Figure 2) to teach it how valid joins look. We explored multiple alternatives to the random forest model in Appendix C.1 below, but no method stood out as a universal best choice.

## A.2    Dynamic threshold behaviour

One of the practical limitations of the presentation in Algorithm 1 is that the validation loop as presented would keep running until enough synthetic samples have been admitted. This results in an infinite loop if there are no more plausible joins to be made, or if the threshold of acceptance, $\theta$, has been set too high.

Two practical remedies described in section 3 are a maximum number of iterations (max_iter = 100) and to oversample the partition synthetic datasets $s_p$'s such that there are more opportunities for valid joins to be made. In most of our experiments, using a multiplier of 4 to the size of the training data was enough to achieve a dataset of sufficient size before the maximum number of iterations was reached.

Additionally, we added the option for setting the acceptance threshold of the validator model dynamically. Both to be set automatically in the first iteration, to accept a certain percentage of joins (e.g., top 10%), and also to lower the threshold slightly if an iteration did not admit any queries during a validation round. This ensures that mainly the best-looking queried samples are accepted, and that multiple reshuffling rounds are permitted for sub-optimal combinations. There is certainly the danger that this behaviour could promote overfitting or increase "selection-bias" in the admitted samples; however, we found the differences between

---

[6]The repository linked in the Introduction (Section 1) holds the implementation, tutorial, and codebooks to reproduce the experimental results.

using the dynamic and tied-down behaviours to be insignificant and minor ($\lesssim 5\%$), in favour of the dynamic case. Because the dynamic behaviour is more reliable in achieving a dataset of sufficient size, we use this option in most of our experiments. In Appendix C.3 we explore setting a static threshold at various levels, and what this means for different validator model quality levels.

## B Comparison to noise injection

While we are considering random partitioning and post hoc concatenation joining we do in Section 4.1, it may pose a worthwhile baseline to consider other forms of noise injection that can be applied to data prior to synthetic data generation. The results shown in Figure 3 demonstrate that partitioning introduces a degradation of the base results; this might also be achieved in other ways.

We repeat the experiment series using the base generative models under three types of perturbations applied at varying strengths: (i) column-wise shuffling, (ii) random replacement, and (iii) Gaussian noise, as introduced in Figure 9. Columnwise shuffling entails selecting $\varphi_1 \cdot n$ of the rows, and then shuffle their values independently and column-wise. Random replacement replaces $\varphi_2 \cdot n$ of the column values with the column mode, and Gaussian noise involves replacing the cell values $x$ with $x' = x + \mathcal{N}(0, \varphi_3 \sigma_x)$.

In Figure 10 we present the aggregated results over all datasets by each noise form, at $\varphi_1, \varphi_2, \varphi_3 \in \{0, 0.1, 0.25, 0.5, 0.9\}$. The granular results are available in the supplement. We show reference lines corresponding to the $\{2, 3, 4\}$ partition results from the main text. The results of these high level averages show, that random partitioning with concatenation joining places about the same on statistical fidelity (PCA and correlation matrix) to synthetic datasets generated at intermediate noise levels, generally outperforms random replacement and random shuffling on the down-stream prediction tasks (hold-out F1 and AUROC difference), and settles at low values for privacy metrics without downright destroying the dataset integrity.

One slightly misleading aspect in this figure is that we plotted the three noise forms together, although the noise degree parameters "$\varphi_1, \varphi_2, \varphi_3$" are generally not directly analogous. However, this presentation still serves to establish that, while these noise forms can achieve similar values to the partitioning case when looking at each metric in isolation, the full consequence of partitioning is not directly covered by any of these cases.

We emphasise that this comparison is intentionally limited to the random partitioning and post hoc concatenation joining setting considered in Section 4.1. The subsequent components of our framework rely on an explicit partition structure, including the validator mechanism and cross-partition dependency recovery, which are fundamentally defined around the existence of discrete partitions.

As a consequence, directly extending a noise-based alternative beyond this initial stage would require designing a substantially different pipeline, including the selection of optimal noise schedules and the introduction of post hoc denoising or dependency recovery mechanisms. Such an exercise would not constitute a controlled baseline comparison, but rather a separate methodological contribution in its own right.

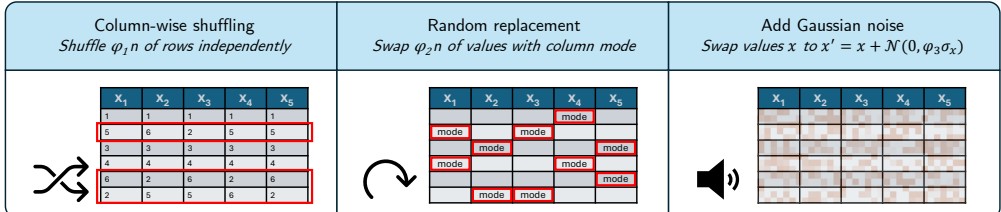

Figure 9: Overview of noise injection comparisons. The figure presents a conceptual overview of the three types of noise injection explored in this appendix. Column-wise shuffling, where a number of rows have their values shuffled between them, Random replacement where values are randomly replaced with the column mode, and Gaussian noise where the values directly have noise added to them.

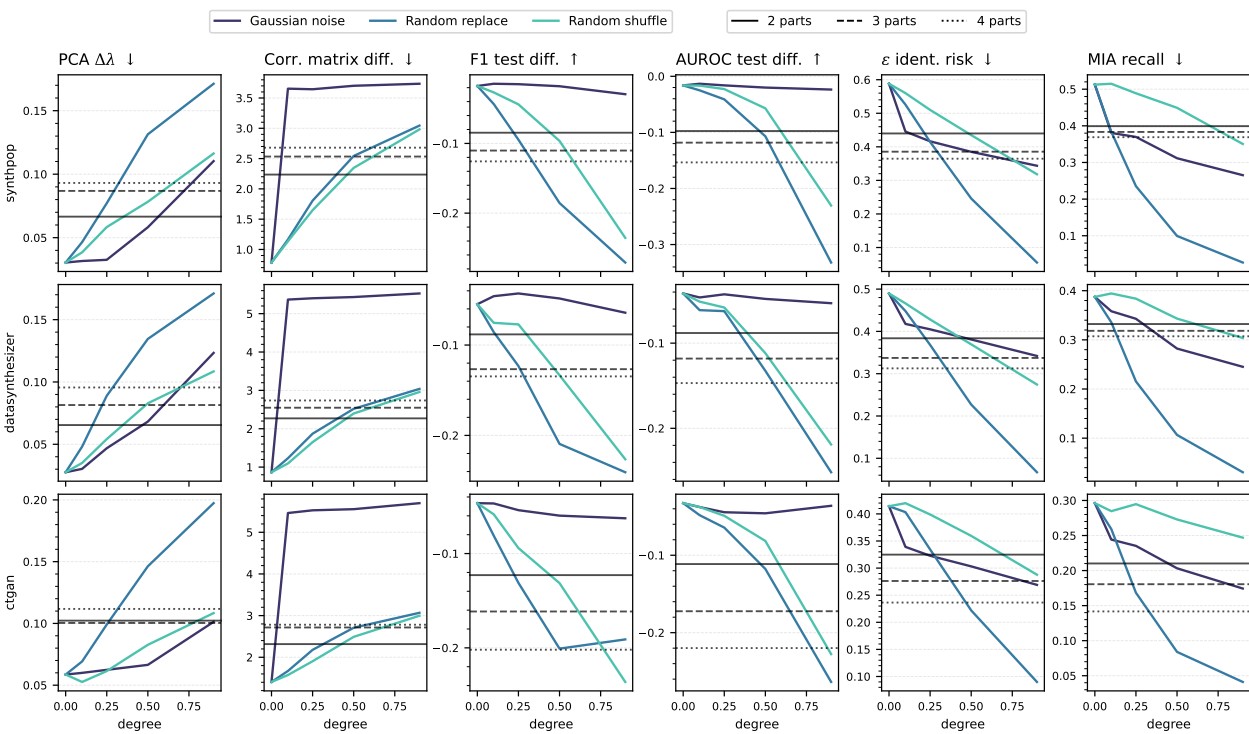

Figure 10: Evaluation metrics vs. noise degree. The figure shows high level results of noise injection (average over datasets, average over repetitions; confidence intervals are left out for visual clarity), with reference lines for comparison with the partitioning results from Figure 3. It is difficult to say anything conclusive based on these results.

For this reason, in the remainder of our experimental evaluation, we instead compare against fully specified and well-established generative frameworks, namely TVAE (Xu et al., 2019) and TabDDPM (Kotelnikov et al., 2023), which natively incorporate noise, compression, and denoising within a coherent end-to-end methodology. These models provide a principled reference for understanding how information degradation and recovery are handled in mature generative pipelines, and allow for a fairer and more informative comparison than ad hoc extensions of noise-based perturbations.

## C    Additional experiments

This appendix holds additional experiments, mainly concerning how we selected and optimised the validator model for the main experiments. The central message of the main text is that partitioning and post hoc joining seem worthwhile to explore for balancing utility and privacy in tabular synthetic data generation; relatively naïve choices appear to deliver promising results, particularly for mixed-model generation. However, how to best choose partitions, models, and how to best do joining are not fully answered in this work.

The following experiments were left as an appendix, since the validator model is, after all, just one way of doing the joining operation in the DGMs framework, albeit one that comes with almost as many new questions to explore.

### C.1    Using different backends for the joining validator

In the main text, we considered the number of partitions, concatenation vs. validation and mixing models as methods for increasing privacy of synthetic data with disjoint generation. In our demonstrations, we have been using a random forest classifier as backend; this is, however, not the only viable choice for the validator

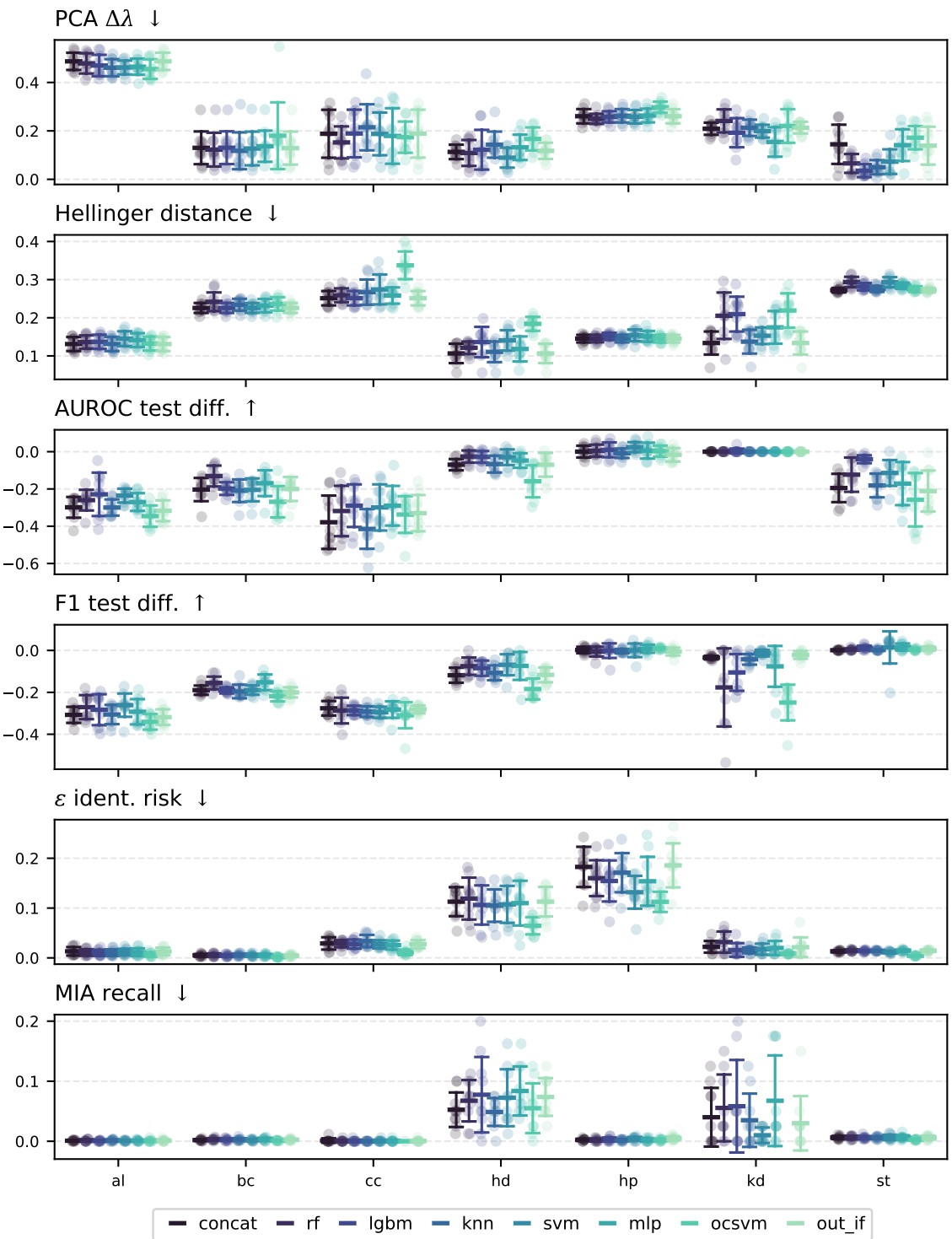

Figure 11: Effect of different validators. This figure shows the effect of using different validator models on the results for different datasets using the synthpop-DPGAN DGM. The average and standard deviation of 10x repeated individual experiments are shown; the individual measurements are also indicated with faint markers. There are differences between validators, but also no consistent discernible patterns across all datasets.

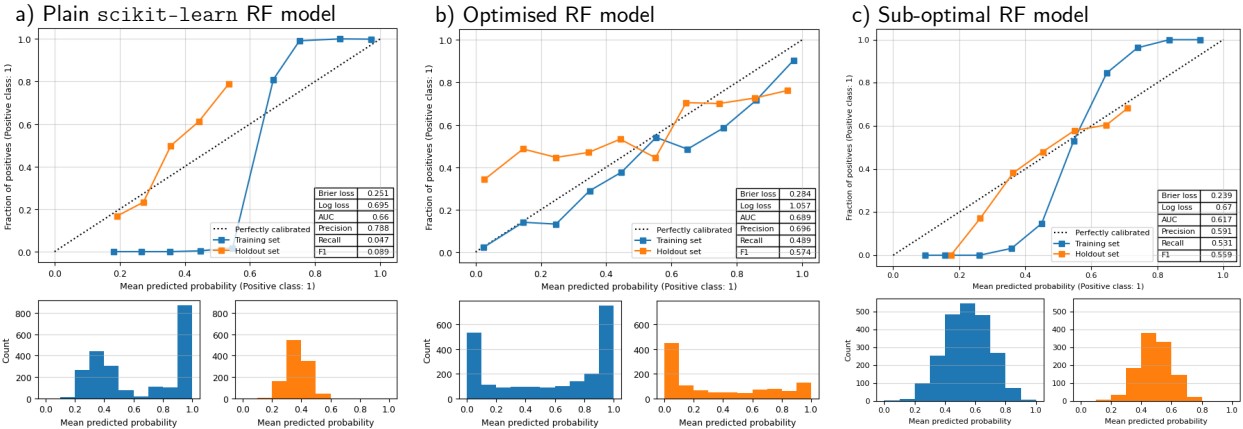

Figure 12: Calibration plots showing optimisation in effect. The three calibration displays show the difference between a model with no optimisation, with optimisation and two with suboptimal optimisation. All are random forest models from `scikit-learn`. The line plots show both predicted probabilities of training and test (holdout) data. The histograms below show how many samples each of the points in the curves above represents: the ideal distribution is "U"-shaped. The metrics results shown are measured for the holdout set.

model. In this appendix, we show how a selection of other common classification models perform, and we also experiment with outlier detection models and one-class classification.

To assess how different validator models affect the quality of the results, we experiment with using LightGBM (LGBM), KNN, SVM, and MLP as validator models compared to concatenation and the random forest model seen throughout this study. We also experimented with a one-class classifier and an outlier detection model (each working a little differently from the presentation in Algorithm 1; see the implementation for details). We employ the synthpop-DPGAN DGM, with the correlation optimised partitioning and perform 10 measurements for each of the regular datasets in Table 1 with every validation model.

The results seen in Figure 11 show that the choice of validator model can be quite impactful on the performance of a synthetic dataset on the presented metrics. Unfortunately, however, there are no clear, consistent patterns; only perhaps that variance seems more pronounced for datasets with fewer records (i.e., `cc`, `hd`, `hp`, and `kd`). Most of the time, the validator models are closely grouped (and that includes concatenation joining), but on occasion, one or two models fall outside of the grouping for a single dataset, for better or worse. For example, the one-class SVM for the heart disease (`hd`) dataset, or the significant variance of the random forest model on the F1 difference metric for the `kd` dataset. We can only say that trying out a variety of validator models for a particular dataset and DGM may yield different benefits/complications. Perhaps our choice of the random forest classifier is not too unreasonable as a first voyage into DGMs. However, more work is needed to determine if any one validator model type is preferable in any particular scenario.

## C.2 Validator effectiveness, hyperparameters optimisation, and calibration

The presented experiments have many "moving parts", and therefore the results have many sources of variability. One that should clearly be investigated was the fit quality of the validator model. For example, a model that learns too well how to distinguish valid and invalid joins in the training set will likely not perform well out of sample. A validator model that underfits, on the other hand, will not pick up on the instrumental cross-correlation and thus not know the difference between plausible and implausible joins any better than random concatenation; perhaps worse due to spurious biases introduced. Indeed, choosing a good validator model is not all down to selecting the most effective discriminator architecture.

In all of our experiments, the validators presented had hyperparameter optimisation and output calibration steps applied to them. An example of a plain `scikit-learn` random forest classifier with no optimisation is

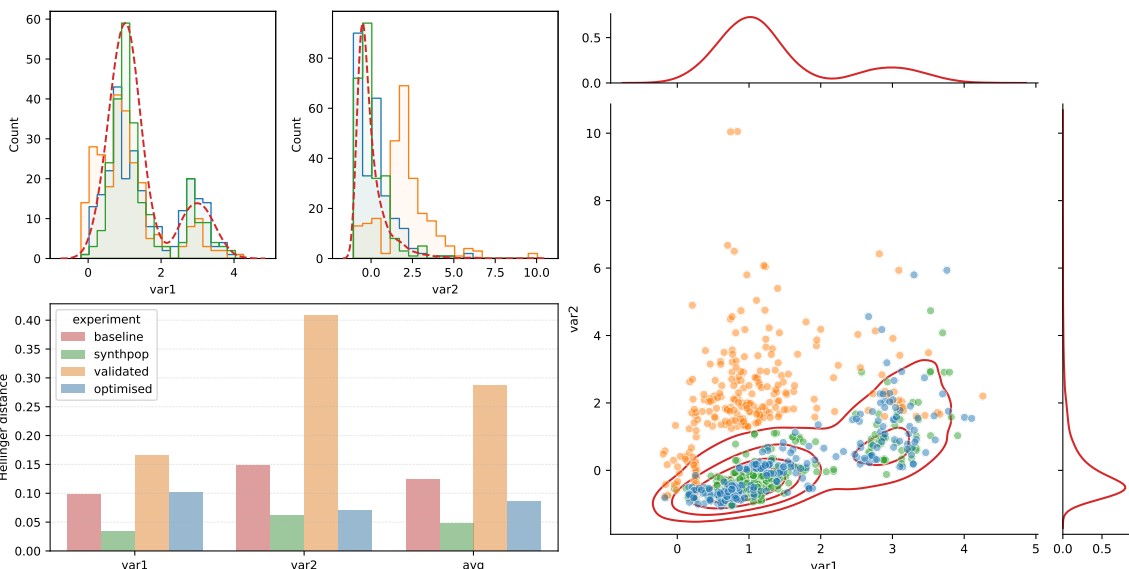

Figure 13: Impact of validation on distributional similarity. The figures demonstrate how using validation joining can disrupt the marginal and joint distribution structures in a dummy dataset. Optimising the hyperparameters of the joining validator and using calibration does apparently help to mitigate the undesirable behaviour. The Hellinger distance is measured to a sample of size 250 from the ground truth distribution, the "baseline" is a *different* sample of 250 items from the ground truth distribution.

shown in Figure 12a for joining the Hepatitis dataset on the correlated partitions. This model seems rather effective at distinguishing the authentic joins to the point where it almost overfits to them. On the other hand, for the "wrong joins" it assigns almost a bell-curve probability distribution, suggesting that perhaps some of them appear more plausible than others. Figure 12b shows the effect of our optimisation steps, which significantly alter the distribution of predicted probabilities, making the model more confident and applicable to the holdout data. We remark that across all datasets the optimised parameter choices for the random forest classifier were typically more cut back than with the default parameters, e.g., fewer estimators than by default (5–20), and with a cap on the tree depth (10–15). To exemplify what it means to have suboptimal optimisation, we also created a joining validator model deprived of resources in the optimisation. This resulted in the model presented in Figure 12c, which is severely underconfident, identifying only a tiny amount of training samples with certainty. Underconfidence leads to high error rates on both types of samples, including those in the holdout data. While some classification error is expected in this scenario due to random bad joins accidentally looking plausible, and likewise, some actual records being out of the known distribution to an extent that they are deemed improbable, the optimised model shows us a much more passable performance and believable consistency.

As an example of how improper care for optimisation and calibration can affect the marginal and joint distributions of data, we present a small experiment on dummy data. In Figure 13, we show marginal and joint distributions of variables "var1" and "var2". We use 250 samples from the ground truth distribution as "training data" and measure the Hellinger distance to it from "synthetic" samples. As a baseline, we use another sample of 250 from the ground truth. We also train a `synthpop` model as a point of comparison, and then we construct two DGMs, one with a standard `scikit-learn` random forest classifier, and another that is subjected to hyperparameter optimisation and calibration. We use the same true samples from the ground truth marginal distributions as "outputs" from the generative models, so that the observed signal can be unambiguously attributed to the decisions of the validator models. The results show that the unoptimised validator model can take otherwise perfect outputs from the "generative models" and entirely destroy the resemblance of both the marginal distributions and the joint output space. In contrast, we show that a well-optimised validator model can produce more sound results.

## C.3 Static acceptance threshold setting

In all of the experiments we have been using the dynamic threshold system described in Section A.2, which automatically sets the threshold to accept the top 10% of the first wave queries (in Figure 12, this would correspond to the thresholds landing at roughly 0.45, 0.8, and 0.6 of the orange distribution). Looking at the spectra of predicted probabilities in Figure 12 raises another question: what is the difference between those items that the model attributes a 0.8 probability and those that are placed at 0.4? We can speculate that the query items are sorted according to some perceived authenticity detected by the model, but we do not *actually* know. If the validator model has no notion of authenticity for new, never-before-seen samples, then the apparently improved privacy could stem from a mode collapse introduced during the joining.

We can investigate if this is indeed what happens by a straightforward experiment: We can enforce a strict threshold for accepting candidate joins at various levels along the calibration curve rather than setting it based on a percentage of the best queries. If we set the threshold all the way to the left at $\sim 0\%$, all queries would be accepted, as when concatenation is used for joining. Increasing the threshold should reveal whether the bias in the admitted samples is acceptable or detrimental.

The results of this experiment are shown in Figure 14. There are several noteworthy behaviours to consider: The plain `scikit-learn` model and the suboptimal model exhibit little change before the threshold reaches approximately halfway through the bell curves in their probability spectrum. On the other side, the performance on most utility metrics suddenly deteriorates, with some improvement to privacy, indicating the expected mode collapse (privacy can be improved due to overfitting to only a few samples as opposed to widespread dataset violations). Hence, the increased threshold amplifies the selection bias with which samples are admitted. The optimised model separates itself from the others early on (by leaving behind a lot of low-valued samples) and then steadily improves on correlation, epsilon risk, and the DCR metric. The other metrics are slightly worsened, but nowhere to the same extent that we see for the other two validator models. The PCA eigenvalue metric does not suggest any significant differences, although a slight inclination may be debated.[7]

Evidently, our optimised model could be better, but at least it does not suffer from the sudden sharp drops/climbs exhibited by the other models. That said, and while the differences between the models at the ends of their trajectories seem to be significant, the size of the numbers on the y-axis is, for the most part, not hugely impactful. We are hesitant to suggest using sub-optimal models for improving privacy (not knowing what sort of biases we might introduce), but in principle, we expect that one could find a validator model and optimise hyperparameters to balance utility and privacy in accordance with various specifications. Checking if the validator model is severely biased would be a crucial step to make this approach work ethically.

# D Evaluation metrics

In this appendix, we briefly explain each of the metrics used in the paper. For further details, we refer the reader to the SynthEval paper (Lautrup et al., 2024b), which includes references and implementation details of most of the metrics.

## D.1 Utility evaluation

PCA *eigenvector difference* and *eigenvector angle difference* are two recent metrics that propose that the projection of real and synthetic samples should be similar. It measures both the difference in eigenvalues and the angle between the first principal component vectors which makes two separate measures that quantify populational alignment of the synthetic data (Rajabinasab et al., 2025). Both metrics are best when closer to zero. *Hellinger distance* is a univariate metric that measures the similarity of the real and synthetic marginal distributions as a value between 0 and 1, where closer to zero implies better similarity (Le Cam & Lo Yang, 2000). The full statistic is an average across all variables. *Correlation matrix difference* computes the pairwise correlation matrix in the real and synthetic variables and subtracts them. If they are similar, the difference

---

[7]The plots for the ML-metrics and MIA risk are shown in the code supplement. They are chaotic, and there are no noteworthy patterns or discernible effects worth mentioning for these metrics.

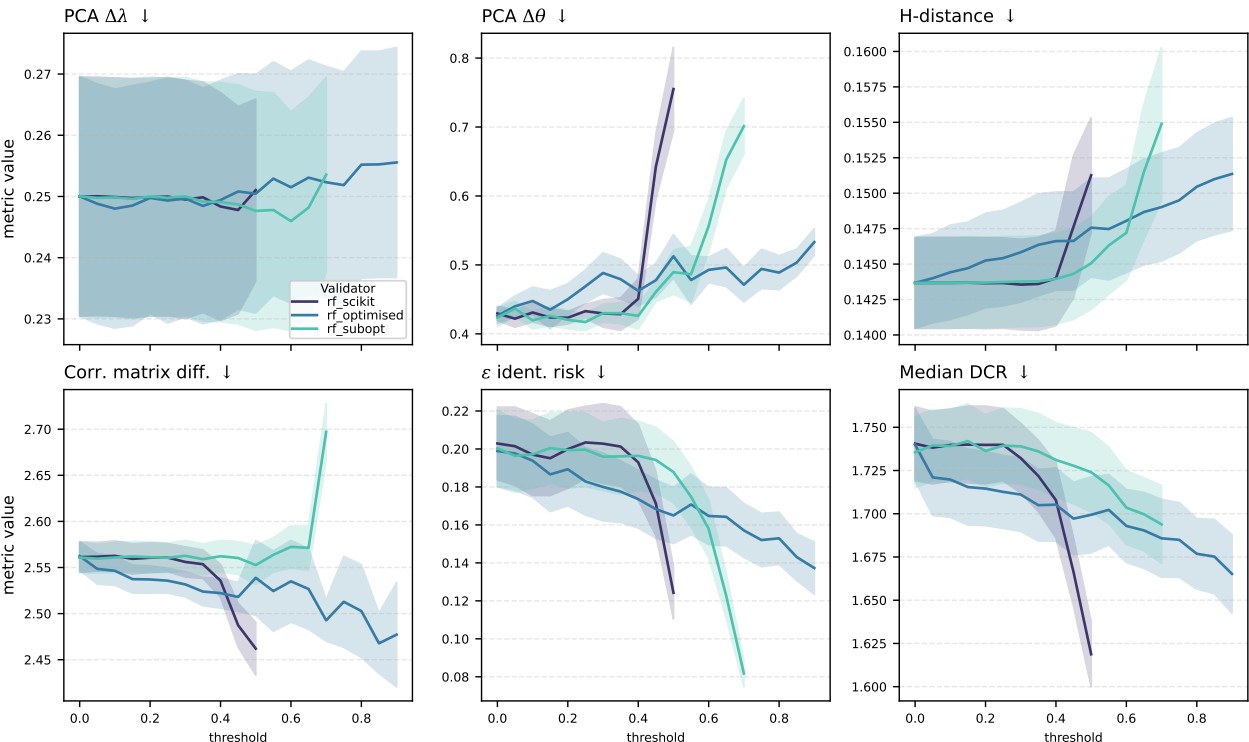

Figure 14: Evaluation metrics of datasets built at different validator thresholds. The line graphs show the result of evaluating synthetic datasets assembled using the three aforementioned validator models at different static thresholds. The confidence bounds are errors for 10x repeated experiments. The models are not run all the way to completion due to a deficit of accepted samples.

map is close to zero, the Frobenius norm is taken to get this to a single value. In SynthEval (Lautrup et al., 2024b), the correlations between categoricals are treated with Cramer's V, and the categorical-numerical correlations using correlation ratio $\eta$. *AUROC difference* and *accuracy difference* measure the practical usability of the synthetic data in predicting a target variable in new authentic records by training predictive models on the real and synthetic data. For AUROC, the difference in performance on a holdout (one that was not used for training the generative model) set is computed. For accuracy difference, both the accuracy of 5-fold cross-validation and holdout sets are considered for four different classification models, and the overall difference between trained on real vs. synthetic data is computed. The results have a sign to denote if the synthetic data are worse (negative) or better (positive) than using the real dataset for training a classification model.

For the sake (AUROC diff) that relies on a binary outcome variable, we had to binarise the outcome column of the "hepatitis" (`hp`) dataset.

## D.2 Privacy evaluation

*ε-identifiability risk* is defined as the fraction of synthetic data points that are "too-close" to real records, measured by the column-entropy weighted distances between real-and-synthetic vs. real-and-real data points (Yoon et al., 2020). *Distance to closest record* is another popular distance-based metric, in this study we use the distance-normalised median DCR to avoid some ambiguities that sometimes happen with average DCR. Finally, we also consider the *precision and recall* of a worst-case-assumptions adversarial attack to infer membership. This attack model assumes that an adversary has access to some full real records (represented by a mix of training and holdout samples). Using the synthetic data, a model is trained to identify if a query

sample was used for training the generative model or not (Emam et al., 2022). The recall is the fraction of correct samples retrieved, and the precision is the confidence with which they are identified.

