# OpenReview forum: "Disjoint Generation of Synthetic Data"
_TMLR — Accepted by TMLR_

### Review · Reviewer_Wq8u · 2026-03-03

**Summary Of Contributions:**

The authors introduce a general framework for synthetic tabular data generation named Disjoint Generative Models (DGMs), and present and analyse a few concrete instantiations of the approach. In general, DGMs operate by partitioning the variables in a tabular dataset into disjoint subsets, individually modelling each subset with a generative model, generating partial samples from each generative model, and rejoining samples via a joining operation to arrive at synthetic full samples. Within this high-level framework, there are multiple design choices, namely how partitions are chosen, which generative models are used, and how rejoining is done. The authors investigate specific choices for each of these considerations. They assess both random partitioning and partitioning that maximises or minimises the ratio of external to internal variable correlations. They also examine multiple generative models. Finally, they consider rejoining via random concatenation, or using a validator model to reject unlikely samples. They present empirical results on 8 standard tabular datasets, assessing the produced synthetic data in terms of statistical similarity, downstream utility, and privacy.

**Strengths**
1. The idea behind DGMs is intuitive
2. The manuscript is very well written. It is easy to follow and comprehensive in its explanations.
3. The methodology behind DGMs is clearly explained, including useful figures and concrete algorithms to support the reader's understanding. The supplied codebase is well documented and includes nice tutorials. This further helps understanding, and facilitates reproducibility.
4. The empirical evaluation is relatively comprehensive, with 8 datasets being used with varying sizes/compositions. Experimental takeaways tend to hold across the majority of the data, and across multiple base generative models.

**Weaknesses**
1. It would be good to see some further baseline comparisons with more recent generative models, which may offer better privacy-utility trade-offs by better modelling the high-dimensional distributions considered without overfitting. Especially when making the claim of 'state-of-the-art' performance for downstream accuracy/AUC, as in the abstract, this requires verification by testing against a broader set of modern models (e.g. diffusion models, VAEs, etc.).

**Additional Comments:**

n/a

**Audience:**

Yes

**Audience Explanation:**

Synthetic data generation is a topic in machine learning that currently enjoys a high level of engagement, given the volume of publications in this area. This paper offers a good contribution in this field, and will be of interest to the TMLR community. The approach is relatively novel, and concrete recommendations for which design choices work best in specific scenarios and to achieve desired outcomes add to its practical utility.

**Claims And Evidence:**

Yes

**Claims Explanation:**

The claims made, and the corresponding parts of the paper that validate them, are:
1. DGMs lead to increased privacy. This is clearly demonstrated in Figure 3, as empirical privacy metrics improve with the degree of partitioning.
2. DGMs lead to faster training. This is clearly demonstrated in Figure 5, as wall-clock time reduces with the degree of partitioning.
3. DGMs enable training with a mixture of generative models, to achieve a better privacy-utility trade-off than the base models. This is clearly demonstrated in Table 3, as the mixture model using synthpop and DPGAN achieves the best result. However, this claim is overstated in the abstract, where the claim that mixed-model synthesis leads to SOTA performance requires more evidence.

**Requested Changes:**

- (critical) Include some more baseline methods to validate the claim for SOTA accuracy/AUC in the abstract, or modify the claim.  This would not necessarily require comprehensive inclusion of more generative models within DGM set-ups, but just as baselines that model the entire data distribution at once, to assess how their privacy-utility trade-offs compare with the proposed DGM architectures. These could include TabDiff [1], TVAE [2], ARF [3], which all perform well on recent tabular synthetic data benchmarking [4].

[1] Shi, Juntong, et al. "TabDiff: a Mixed-type Diffusion Model for Tabular Data Generation." The Thirteenth International Conference on Learning Representations.

[2] Xu, Lei, et al. "Modeling tabular data using conditional gan." Advances in neural information processing systems 32 (2019).

[3] Watson, David S., et al. "Adversarial random forests for density estimation and generative modeling." International Conference on Artificial Intelligence and Statistics. PMLR, 2023.

[4] Jiang, Xiangjian, Nikola Simidjievski, and Mateja Jamnik. "TabStruct: Measuring Structural Fidelity of Tabular Data." ICLR 2026.

---

> ### Author Response · Authors · 2026-04-15
> **Response to Wq8u**
>
> We thank the reviewer for the thorough reading and constructive suggestions. In response, we have clarified the scope of the paper and added additional baseline comparisons, which we believe improve the clarity and completeness of the work.
>
> > RC1: Include some more baseline methods to validate the claim for SOTA accuracy/AUC in the abstract, or modify the claim. This would not necessarily require comprehensive inclusion of more generative models within DGM set-ups, but just as baselines that model the entire data distribution at once, to assess how their privacy-utility trade-offs compare with the proposed DGM architectures. These could include TabDiff (Shi et al. 2024), TVAE (Xu et al. 2019), ARF (Watson et al. 2023), which all perform well on recent tabular synthetic data benchmarking (Jiang et al. 2026).
>
> Thanks for pointing this out, and for suggesting comparison points and references. We have added additional baselines (CTGAN, TVAE, ARF, ADS-GAN, and TabDDPM) in the mixed-model results Section 4.5.
>
> - See end of second and third paragraphs, Table 3, the “other datasets” paragraph, and Figure 8.
>
> We find that these additions make the performance look stronger; however, we did not mean to claim general state-of-the-art performance but rather meant to write that “we get utility comparable to that of established ‘high-utility’ models, while having privacy metrics come out as comparable to ‘high-privacy’ models”. We have softened the claim in the abstract for better balance, but ask for advice if this was the correct decision?
>
> -	Changed “state-of-the-art” to “highly competitive”
>
> We tried to make TabDiff work, but it was not easily incorporated into the existing codebase, which use Synthcity for many of the deep learning models. The other reviewer suggested TabDDPM, which is available with Synthcity: in the (Jiang et al. 2026) paper you shared, the two diffusion models seem fairly competitive (on 95% confidence intervals), so we hope that this inclusion is a fair compromise?
>
> We added the references (Xu et al. 2019), (Watson et al. 2023), and (Jiang et al. 2026) to appropriate passages of the paper. Thanks for these!

---

### Review · Reviewer_wJCc · 2026-03-09

**Summary Of Contributions:**

The paper proposes Disjoint Generative Models, an algorithm for generating tabular datasets. They key contribution is to partition tabular datasets by column and then train separate generative models on each partition. The outputs are combined through a joining operation using a validator model (trained on real data), that assess the quality of these joining operations. The claims made by the authors are that this can improve empirical privacy metrics while maintaining utility.

**Audience:**

No

**Audience Explanation:**

The key insights provided in the paper are:
4.1: Partitioning improves privacy but harms utility: Partitioning is a form of noise induced in the generative process by fragmenting information. This will harm utility for sure and provide empirical privacy gains. There are N number of operations that can be applied to the tables before doing the generation such as adding noise : replacing values randomly with mode, shuffling values randomly across rows etc. For columns with numerical values again, cell level noise could be added etc What is the impact of performance under the influence of all these types of noise and what makes partitioning the best is not clear.

4.3: Partitioning improves efficiency of some models: As the authors say in the paper, this is not a surprising result.

4.4 It matters which variables end up in which partition: Again, when inter-partitions relationship do not need to be modeled, this becomes easy for the join operation, if not, it becomes a task for the validator.

These claims are trivial for the problem statement and the solution.

**Broader Impact Concerns:**

There are no ethical concerns in this work.

**Claims And Evidence:**

No

**Claims Explanation:**

1. The overall framework proposed lacks any formal privacy guarantees. While it has been called out in section 4.5, the value add of the proposed method and claims made are very limited.
2. The baseline comparisons are all done on the non-disjoint version of the same models which does not make the proposed method "state-of-the art". For instance, how does this method compare with  TabDDPM ((https://proceedings.mlr.press/v202/kotelnikov23a/kotelnikov23a.pdf)

**Requested Changes:**

Experiments with other forms of noise injection in tables (Critical to show why partioning is the best way)

---

> ### Author Response · Authors · 2026-04-15
> **Response to wJCc (part 1/2)**
>
> Thank you for taking the time to engage with our manuscript and for offering suggestions for improvement. Overall, while we have incorporated changes where appropriate, we believe the manuscript’s core framing and contributions remain well aligned with the journal’s scope.
>
> > Claims: The overall framework proposed lacks any formal privacy guarantees. While it has been called out in section 4.5, the value add of the proposed method and claims made are very limited.
>
> Thank you for highlighting the importance of formal privacy guarantees.
>
> First, we would like to clarify that we do not claim that the proposed framework provides formal privacy guarantees. We have tried as best as we could to establish that we only report empirical heuristic privacy. See for example: last sentence in abstract *“(…) significantly lowering the empirical re-identification risk.”*, last sentence in the second paragraph in the introduction *“(…) balance utility and empirical privacy effectively”*, second paragraph in background which establishes data-centric privacy, and the “please note” disclaimer in the beginning of section 4.
>
> While we agree that differential privacy is an important and widely adopted paradigm, it is also not the only way privacy is being treated within the field of synthetic data generation. Many real-world applications rely on empirical privacy assessments, utility–privacy trade-offs, or domain-specific constraints that are not easily captured within a fully formal framework. Our work proposes disjoint generative models as a broader design space rather than a formal privacy framework.
>
> To acknowledge that formal privacy guarantees are central to many use cases, and that we see it as an important and concrete direction for extending this work, we have added an additional point to the future directions:
>
> - *“Establishing end-to-end privacy, by using differentially private constituent models, validation process, and covariance estimation (Amin et al., 2019), while accounting for the privacy budget across the full framework, could widen the appeal of disjoint generation in more regulated settings and for continual release scenarios.”*
>
> We have also added a broader impact statement where we also stress that
>
> - *“However, as we acknowledge, the privacy we treat here is heuristic rather than formal and is therefore limited to the synthetic datasets themselves rather than the models that produce them. In other words, the framework in its current state does not provide end-to-end privacy guarantees, and the reported reductions in identification risk may not hold in realistic adversarial scenarios.”*
>
> We would also welcome any suggestions for clarifying this distinction or for relevant references to include.
>
> > Claims: The baseline comparisons are all done on the non-disjoint version of the same models which does not make the proposed method "state-of-the art". For instance, how does this method compare with TabDDPM (Kotelnikov et al., 2023).
>
> Thank you for this suggestion. Yes, we rewrote that particular sentence in the abstract many times before settling on the current version, and based on the other reviewer response, this reads as us claiming state-of-the-art without the proper baselines. What we meant to write was that “we get utility comparable to that of ‘high-utility’ models, while having privacy metrics come out as comparable to ‘high-privacy’ models”.
>
> To address this feedback, we have revised the manuscript in the following ways:
>
> -	Soften claims by changing “state-of-the-art” in the abstract to “highly competitive”.
> -	Added additional baseline comparisons in Section 4.5; CTGAN, TVAE, ARF, ADS-GAN and TabDDPM. See in particular the new Figure 8.

---

> > ### Author Response · Authors · 2026-04-15
> > **Response to wJCc (part 2/2)**
> >
> > > Worthwhile findings: (paraphrase) 4.1, 4.3, 4.4, These claims are trivial for the problem statement and the solution.
> >
> > In our opinion, while some of the observed behaviours may appear simple or intuitive at the first glance, they cannot substitute a comprehensive empirical validation just by intuition alone, especially for the first study on a novel method. We believe that on the contrary, these findings are informative and crucial for a comprehensive empirical validation of a novel method.
> >
> > The purpose of the experiments referenced, is not to claim surprising effects in isolation, but rather to systematically, (4.1) establish that even the most naïve thing you can do with the framework has an effect, (4.3) demonstrate that e.g. Bayesian network models can be made applicable to high dimensional datasets, and (4.4) provide empirical groundwork for conditions under which the validator model may be successful. Establishing that these components behave in line with expectations is a prerequisite for interpreting downstream analyses and for enabling future work to build upon the framework with confidence.
> >
> > Moreover, this critique focuses on these three experiments that primarily establish expected behaviour but omits the more interesting findings of the paper (Section 4.5), which demonstrate that disjoint generation with mixed-models combining a high-utility model (synthpop) with a high-privacy model (DPGAN), can achieve a better utility/privacy trade-off than either model alone. This result is neither obvious nor implied by prior intuition, and it directly arises from the interaction of the components introduced in the proposed framework, and we expect that it will be of interest to others.
> >
> > > RC1: Experiments with other forms of noise injection in tables (Critical to show why partioning is the best way)
> >
> > Thank you for this suggestion, we agree it is a reasonable point. We view this comparison as a useful additional reference for understanding the role of partitioning in the proposed framework.
> >
> > In response, we have conducted additional experiments following the setup in Section 4.1 (random partitioning and concatenation), replacing our partitioning procedure with alternative forms of noise injection applied prior to generation. These results are reported in Appendix B, and the appendix is explicitly referenced from the main text to preserve the narrative flow (Section 4 overview and Section 4.1)
> >
> > In the appendix, we present experiments using base generative models under three types of perturbations applied at varying strengths: (i) random row-wise shuffling, (ii) random replacement with mode, and (iii) cell-level Gaussian noise. As discussed there, a key challenge is that noise magnitudes are inherently incomparable across these types of perturbation, and by extension, with the number of partitions. Moreover, the downstream metrics respond differently to different noise mechanisms: for example, Gaussian noise at the applied levels has minimal impact on predictive performance while substantially disrupting correlation structure, whereas other perturbations preserve correlations at much higher noise rates.
> >
> > The primary conclusion from this comparison is therefore qualitative: similar effects to partitioning can be achieved via noise injection, but there is no single noise type or strength that reproduces the behaviour of partitioning across metrics. We include this analysis as evidence that partitioning can be understood as one structured form of information degradation, rather than as a unique effect.
> >
> > Finally, while noise injections can produce qualitatively similar effects to partitioning, the remainder of our framework is inherently defined around an explicit partition structure. Extending the noise-based approach through the full pipeline would therefore require designing and validating a different methodological framework, rather than constituting a direct or fair baseline. We hope that the additional baselines we added with TVAE and TabDDPM in later Section 4.5 can serve to further contextualise how noise, information compression, and denoising behave within established generative frameworks.

---

### Review · Reviewer_Femj · 2026-04-08

**Summary Of Contributions:**

This paper proposes Disjoint Generative Models (DGMs), a framework for tabular synthetic data generation that partitions a dataset column-wise, trains separate generative models on each partition, and reassembles the outputs via a joining operation, often involving a 'validator' model to recover inter-partition dependencies. The central idea is a natural application of divide-and-conquer to generative modelling. The paper demonstrates three main benefits: improved empirical privacy, computational efficiency for certain model types, and the ability to mix generative models across partitions. The mixed-model result combined a high-utility model (synthpop) with a differentially private model (DPGAN) on different partitions, showing that the framework can achieve a better utility / privacy tradeoff than either model alone. The paper is honest about its limitations and is clearly positioned as an exploratory contribution rather than a definitive solution.

Key strengths:

- Well-motivated framework with a clean conceptual design
- Honest about limitations; does not overclaim
- Mixed-model generation is a practically useful result
- Open-source implementation provided

Key weaknesses:

- Privacy analysis is incomplete: the partition design and validator training steps are not accounted
- Lack of principled guidance for practitioners on how to choose partition schemes and the number of partitions
- Experiments are limited to small datasets and classification-based utility metrics

**Audience:**

Yes

**Audience Explanation:**

The problem of balancing utility and privacy in synthetic tabular data generation is of broad and growing interest to the machine learning community, particularly in healthcare, finance, and policy domains. The DGMs framework is simple enough to be accessible, and the mixed-model result offers a practically actionable insight.  The paper opens a concrete and interesting research direction that the community can build on.

**Broader Impact Concerns:**

The paper lacks a Broader Impact Statement. We recommend adding one to clarify that the reported privacy improvements are empirical only and do not constitute formal guarantees, to avoid misuse in regulated settings such as healthcare or finance.

**Claims And Evidence:**

Yes

**Claims Explanation:**

The core empirical claims are well-supported: the privacy / utility tradeoff under increasing partitions is consistently demonstrated across multiple datasets and model types, and the mixed-model results are replicated across seven datasets with repeated experiments. However, privacy claims should be more carefully scoped.  The partition design and validator training both access real data without protection, meaning the reported improvements reflect empirical risk on the final synthetic output only, not end-to-end privacy preservation.

**Requested Changes:**

Critical:
- Privacy budget accounting for the full pipeline. The current paper evaluates privacy only on the final synthetic dataset, but the partition design step (correlation matrix computation) and the validator training step both access real data without any privacy protection. This directly voids the differential privacy guarantees of constituent models like DPGAN, as the authors briefly note in Limitation 1 but do not fully address.  Ideally, the authors should discuss how differentially private correlation estimation could be incorporated, even if only as a future direction.

- Guidance on partition scheme selection. The paper demonstrates that partition design matters significantly (Section 4.4) but provides no actionable guidance for practitioners facing a new dataset. How should one choose the number of partitions?  Even empirical heuristics derived from the existing experiments would be valuable.

Non-critical but recommended:


- Broader utility evaluation. All utility metrics are ultimately tied to classification performance. It would strengthen the paper to include at least one regression or distributional task to assess whether the cross-partition dependency loss has asymmetric effects depending on the downstream use case.

- Theoretical analysis of validator effectiveness. The paper currently has no formal analysis of when or why the validator can recover cross-partition dependencies. Even a brief discussion of the conditions under which the validator is expected to succeed or fail would make the framework more principled.

---

> ### Author Response · Authors · 2026-04-15
> **Response to Femj (part 1/2)**
>
> Thank you very much for the positive and very encouraging feedback! Below we address all of your comments and have updated the manuscript accordingly to incorporate additional clarifications and results.
>
> > RC1: Privacy budget accounting for the full pipeline. The current paper evaluates privacy only on the final synthetic dataset, but the partition design step (correlation matrix computation) and the validator training step both access real data without any privacy protection. This directly voids the differential privacy guarantees of constituent models like DPGAN, as the authors briefly note in Limitation 1 but do not fully address. Ideally, the authors should discuss how differentially private correlation estimation could be incorporated, even if only as a future direction.
>
> Thank you for highlighting this important point. You are absolutely right that, in its current form, the framework does not provide end-to-end differential privacy, and we hope that it does not come across as us claiming that(?). We do apply models with differential privacy property as parts of our experiments (mainly because the data they produce usually rank well on empirical privacy measures), but in those experiments other models such as the validator model (and indeed also the correlation estimation), access the real data without constraints thus breaking the compositional DP guarantees.
>
> While our current focus is on settings where the generative model is not published and only a single dataset is created and used for research or educational purposes in a regulated environment, we fully agree that formal privacy guarantees are central to many use cases, and we see it as an important and concrete direction for extending this work.
>
> As for your requested change, we have made the following changes and additions:
>
> -	Page 5 disclaimer: *“(…) including all constituent submodels and validation components, remains an open problem and should be explored in future works.”*
> -	Page 13 future directions: *“Establishing end-to-end privacy, by using differentially private constituent models, validation process, and covariance estimation (Amin et al., 2019), while accounting for the privacy budget across the full framework, could widen the appeal of disjoint generation in more regulated settings and for continual release scenarios.”*
> -	Page 14 broader impact statement, paragraph 2: *“However, as we acknowledge, the privacy we treat here is heuristic rather than formal and is therefore limited to the synthetic datasets themselves rather than the models that produce them. In other words, the framework in its current state does not provide end-to-end privacy guarantees, and the reported reductions in identification risk may not hold in realistic adversarial scenarios.”*
>
> We hope that these adjustments meet your concerns. We would very much appreciate your suggestions on how we could further clarify the distinction between empirical privacy and formal DP, or references you believe would be particularly appropriate to cite.
>
> >  RC2: Guidance on partition scheme selection. The paper demonstrates that partition design matters significantly (Section 4.4) but provides no actionable guidance for practitioners facing a new dataset. How should one choose the number of partitions? Even empirical heuristics derived from the existing experiments would be valuable.
>
> Thanks for raising this. Yes, you are right that Section 4.4 is lacking a clear takeaway message. We added the following paragraph to Section 4.4;
>
> *“Based on these findings, it would arguably be practical to ensure a high degree of dependency is shared between the partitions if using validated joins. Alternatively, if using concatenation joining, as little information as possible should be shared among the partitions in order to ensure seamless joining.”*
>
> We mostly worked with fewer partitions to limit the combinatorically growing number of choices for submodels and optimisations that would need to be checked in our experiments. On the high-dimensional dataset example, we experienced that the validator did take slightly longer to finish validation on the highest numbers of partitions (still insignificant compared to the training time) suggesting that higher numbers of partitions make the random joining before the validation less viable (perhaps also noisier and more biased). Recursion or a more intelligent record-matching technique could help here in future work.
> In the future directions, we suggested that it would be nice with an investigation of the partition number vs partition size, and that better techniques for assigning variables and models to partitions could be useful.

---

> > ### Author Response · Authors · 2026-04-15
> > **Response to Femj (part 2/2)**
> >
> > > RC3: Broader utility evaluation. All utility metrics are ultimately tied to classification performance. It would strengthen the paper to include at least one regression or distributional task to assess whether the cross-partition dependency loss has asymmetric effects depending on the downstream use case.
> >
> > Thank you for this thoughtful suggestion. We fully agree that a thorough multifaceted evaluation is important.
> >
> > We would like to clarify that, while all datasets considered in this work are classification datasets, our utility evaluation extends beyond classification performance. In total, we report 12 utility measurements, including several distributional and feature-level metrics (e.g., marginal, pairwise and structural statistics) , which we feel is fairly comprehensive (see for example, the Jiang et al. paper linked by the other reviewer; there they use 10 dimensions in a dedicated benchmark paper). All the metrics we assess are featured in Table 3, but we did calculate them for all the other experiment series. We tried to keep the figures focused on the interesting results and have the remainder in the code supplement. If there is anything in particular you are looking for in any of the experiments, please let us know.
> >
> > In response to your comment on cross-partition dependency loss; in the paper we alluded to spurious biases introduced by the validator model. These mentions refer to the finding that the concatenation case maintains marginal metrics (i.e. Hellinger distance) of the parts, whereas using the joining validator can cause an offset, meaning a distortion of the marginal distributions (which we now show in Figure 6). Moreover, we decided to go back and clean up an old test case we made and added it to Appendix C.2 (how the optimisation of the validator model affects results). This experiment shows how a poorly optimised validator model skews both the marginal and joint distribution, whereas for a well-optimised validator, this effect is largely mitigated. We hope that this experiment may offer some additional insight.
> >
> > > RC4: Theoretical analysis of validator effectiveness. The paper currently has no formal analysis of when or why the validator can recover cross-partition dependencies. Even a brief discussion of the conditions under which the validator is expected to succeed or fail would make the framework more principled.
> >
> > Thank you for this comment. We agree that a principled understanding of when the validator is expected to succeed or fail is important.
> >
> > We would like to clarify that the paper already contains a qualitative analysis of validator effectiveness. In the main text, we explicitly discuss easy and hard regimes for recovering cross-partition dependencies (section 4.4, i.e., strong correlations vs none), and the entirety of Appendix C is dedicated to practical considerations required to make the validation system work in practice, including optimisation dynamics, calibration, and failure modes.
> >
> > That said, we acknowledge that this discussion is currently framed in an intuitive and empirical manner rather than as a formal theoretical analysis. In response to your comment, we have added “validator effectiveness, (…)” to the title of Appendix C.2, and rewritten the passage introducing the appendix in section 4 so that these details are easier to locate.
> >
> >
> > > Broader Impact Statement. The paper lacks a Broader Impact Statement. We recommend adding one to clarify that the reported privacy improvements are empirical only and do not constitute formal guarantees, to avoid misuse in regulated settings such as healthcare or finance.
> >
> > Thank you for raising this point. We added it in just after the discussion; it covers both the many exciting possibilities going forwards, but also the concern that the privacy improvements we show are merely empirical.
> >
> > *“Current literature on synthetic data generation identifies a gap, or perhaps a disconnect, between ongoing efforts for high utility and privacy in synthetic data, and the proposed disjoint generative models framework offers a quite direct method for marrying the breakthroughs in both areas through mixed-model generation. Moreover, joining the efforts of specialised models and performances could also unlock a new frontier for multi-modal learning (which was not explored in this work), as state-of-the-art, text, image, tabular, and time-series models may be used in tandem with a multimodal classification model to achieve multi-modal data generation.*
> >
> > *However, as we acknowledge, the privacy we treat here is empirical rather than formal and is therefore limited to the synthetic datasets themselves rather than the models that produce them. In other words, the framework in its current state does not provide end-to-end privacy guarantees, and the reported reductions in identification risk may not hold in realistic adversarial scenarios.”*

---

### Author Response · Authors · 2026-06-03
**Camera Ready Submission**

Dear editor and reviewers,

Thank you once again for the constructive feedback and comments, which helped improve the manuscript beyond the first submitted version.

We integrated TabDiff into the mixed-model generation experiments (See, in particular, Table 3, Figure 7, text on pages 10-12 and Appendix A.1). The mixed models with TabDiff perform similarly to the other high-utility model but provide a valuable perspective and elevate the comparison to more recent State-of-the-Art.

Thank you for your consideration.

Sincerely,
The Authors

---

> ### Comment · Action_Editor_HEJH · 2026-06-03
>
> I think there is a misunderstanding; the idea of the reviewer (and myself) is to use tabdfif not only in the second panel of the Table, but also inside your own method (panel C+D in table 3).

---

> > ### Author Response · Authors · 2026-06-04
> > **Official Comment by Authors**
> >
> > Thank you for your comment. That was not clear to us from your decision letter. We will start addressing it right away and get back to you once the update is done.

---

> > ### Author Response · Authors · 2026-06-07
> > **Official Comment by Authors**
> >
> > We managed to integrate TabDiff into the mixed-model experiments. (See, Table 3, Figure 7, text on pages 10-12 and Appendix A.1)

---

### Decision · Action_Editor_HEJH · 2026-05-18

**Recommendation:** Accept with minor revision

**Additional Comments:**

Because this is fundamentally an empirical paper, the bar for comparisons should be high from an empirical angle. While the revised manuscript adds several important baselines, I believe that comparison against TabDiff is necessary, given the status of TabDiff as a current state-of-the-art method for tabular diffusion modeling. Including such a baseline would substantially strengthen the empirical positioning of the work and help contextualize the proposed framework relative to the strongest contemporary approaches. The argument that the TabDiff implementation was not possible off-the-shelf is not convincing; if the authors need more time for that, it is possible to extend the deadline for submitting the revised version.

Regarding the concerns about privacy, I see the value in the empirical analysis and would thus defer a formal privacy analysis to future work. However, this also implies that the bar for the empirical analysis is thus higher

Overall, I find the paper to meet the standard for TMLR as a useful empirical contribution that opens an interesting direction for future research, and I therefore recommend acceptance, **conditional** on the inclusion of a comparison with TabDiff.

**Audience:**

Yes

**Audience Explanation:**

This paper proposes a framework for disjoint generation of synthetic tabular data through partitioned generative modeling and post hoc joining. The contribution is primarily empirical, but the reviewers generally agree that the work studies a practically relevant and underexplored design space in a careful and transparent manner. In particular, the mixed-model generation results are viewed as a meaningful contribution, with relevant insights for the community.

**Claims And Evidence:**

Yes

**Claims Explanation:**

The authors responded constructively to the reviews by clarifying the distinction between empirical privacy and formal privacy guarantees, broadening the experimental comparisons, adding additional discussion and analysis, and softening claims where appropriate.